# Plasma proteomics identify biomarkers predicting Parkinson's disease up to 7 years before symptom onset

Jenny Hällqvist [1,2,13] ✉, Michael Bartl [3,4,13] ✉, Mohammed Dakna[3], Sebastian Schade [5], Paolo Garagnani [6], Maria-Giulia Bacalini[7], Chiara Pirazzini[6], Kailash Bhatia [8], Sebastian Schreglmann [8], Mary Xylaki [3], Sandrina Weber[3], Marielle Ernst[9], Maria-Lucia Muntean[5], Friederike Sixel-Döring[5,10], Claudio Franceschi [6], Ivan Doykov[1], Justyna Śpiewak[1], Héloïse Vinette [1,11], Claudia Trenkwalder[5,12], Wendy E. Heywood [1], Kevin Mills[2,14] & Brit Mollenhauer [3,5,14]

Parkinson's disease is increasingly prevalent. It progresses from the pre-motor stage (characterised by non-motor symptoms like REM sleep behaviour disorder), to the disabling motor stage. We need objective biomarkers for early/pre-motor disease stages to be able to intervene and slow the underlying neurodegenerative process. Here, we validate a targeted multiplexed mass spectrometry assay for blood samples from recently diagnosed motor Parkinson's patients ($n = 99$), pre-motor individuals with isolated REM sleep behaviour disorder (two cohorts: $n = 18$ and $n = 54$ longitudinally), and healthy controls ($n = 36$). Our machine-learning model accurately identifies all Parkinson patients and classifies 79% of the pre-motor individuals up to 7 years before motor onset by analysing the expression of eight proteins—Granulin precursor, Mannan-binding-lectin-serine-peptidase-2, Endoplasmatic-reticulum-chaperone-BiP, Prostaglaindin-H2-D-isomaerase, Interceullular-adhesion-molecule-1, Complement C3, Dickkopf-WNT-signalling pathway-inhibitor-3, and Plasma-protease-C1-inhibitor. Many of these biomarkers correlate with symptom severity. This specific blood panel indicates molecular events in early stages and could help identify at-risk participants for clinical trials aimed at slowing/preventing motor Parkinson's disease.

Parkinson's disease (PD) is a complex and increasingly prevalent neurodegenerative disease of the central nervous system (CNS). It is clinically characterised by progressive motor and non-motor symptoms that are caused by α-synuclein aggregation predominantly in dopaminergic cells, which leads to Lewy body (LB) formation[1]. The failure of neuroprotective strategies in preventing disease progression is due, in part, to the clinical heterogeneity of the disease—it has several phenotypes—and to the lack of objective biomarker readouts[2]. To facilitate the approval of neuroprotective strategies, governing

agencies and pharmaceutical companies need regulatory pathways that use objectively measurable markers—potential therapeutical targets as well as state and rate biomarkers—directly associated with PD pathophysiology and clinical phenotypes[3].

The recently emerged α-synuclein seed amplification assays (SAA) can identify α-synuclein pathology in vivo and support stratification purposes but still rely on cerebrospinal fluid (CSF) obtained through relatively invasive lumbar punctures[4]. Therefore, this test remains specialised and not readily suitable for large-scale clinical use. As

peripheral fluid biomarkers are less invasive and easier to obtain, they could be used in repeated and long-term monitoring, which is necessary for population-based screenings for upcoming neuroprotective trials. While the only emerged serum biomarker in the last years, axonal marker neurofilament light chain (NfL), increases longitudinally and correlates with motor and cognitive PD progression[5], it is non-specific to the disease process.

Growing data support evidence of PD pathology in the peripheral system, which increases the likelihood of finding a source of matrices for less invasive biomarkers. We know α-synuclein aggregation induces neurodegeneration, which is propagated throughout the CNS. Evidence indicates that additional inflammatory events are an early and potentially initial step in a pathophysiological cascade leading to downstream α-synuclein aggregation that activates the immune system[6]. Inflammatory risk factors in circulating blood (i.e. C-reactive-protein and Interleukin-6 and α-synuclein-specific T-cells) are associated with motor deterioration and cognitive decline in PD[7,8]. These inflammatory blood markers can even be identified in plasma/serum samples of individuals with isolated REM sleep behaviour disorder (iRBD), the early stage of a neuronal synuclein disease (NSD), and the most specific predictor for PD and dementia with Lewy bodies (DLB)[6]. NSD was recently proposed as a biologically defined term, for a spectrum of clinical syndromes, including iRBD, PD and DLB, that follow an integrated clinical staging system of progressing neuronal α-synuclein pathology (NSD-ISS)[9].

In this study, we used mass spectrometry-based proteomic phenotyping to identify a panel of blood biomarkers in early PD. In the initial discovery stage, we analysed samples from a well-characterised cohort of de novo PD patients and healthy controls (HC) who had been subjected to rigorous collection protocols[10]. Using unbiased state-of-the-art mass spectrometry, we identified putatively involved proteins, suggesting an early inflammatory profile in plasma. We thereafter moved on to the validation phase by creating a high-throughput and targeted proteomic assay that was applied to samples from an independent replication cohort, consisting of de novo PD, HC and iRBD patients. Finally, after refining the targeted proteomic panel to include a multiplex of only the biomarkers which were reliably measured, an independent analysis was performed on a larger and independent cohort of longitudinal, high-risk subjects who had been confirmed as iRBD by state-of-the-art video-recorded polysomnography (vPSG), including follow-up sampling of up to 7 years.

In summary, using a panel of eight blood biomarkers identified in a machine-learning approach, we were able to differentiate between PD and HC with a specificity of 100%, and to identify 79% of the iRBD subjects, up to 7 years before the development of either DLB or motor PD (NSD stage 3). Our identified panel of biomarkers significantly advances NSD research by providing potential screening and detection markers for use in the earliest stages of NSD for subject identification/stratification for the upcoming prevention trials.

## Results

### Proteomic discovery phase 0
We performed a bottom-up proteomics analysis of plasma, which had been depleted of the major blood proteins, using two-dimensional in-line liquid chromatography fractionation into ten fractions and label-free mass spectrometric analysis by QTOF MS$^E$. The discovery cohort consisted of ten randomly selected drug-naïve patients with PD and ten matched HC from the de novo Parkinson's disease (DeNoPa[10]) cohort (details can be found in Supplementary Table 1). This analysis identified 1238 proteins when restricting identification to originate from at least one peptide per protein and at least two fragments per peptide. After excluding proteins with less than two unique peptides or with an identification score below a set threshold (see method section below), 895 distinct proteins remained. Of these proteins, 47 were differentially expressed

between the de novo PD and control groups on a nominal significance level of 95%. Pathway analysis suggested enrichment in several inflammatory pathways. Workflow and Results are shown in Fig. 1, and 2 Supplementary Figs. 1, 2.

### Selection of proteins for the targeted proteomic assay
We next developed a validatory, high-throughput and multiplexed, mass spectrometric targeted proteomic assay based on the potential biomarkers identified in the discovery phase. Additional proteins were also included in the assay, several of which had been identified in previous discovery studies of PD, Alzheimer's disease (AD), and ageing[11]. In addition, we also included several known pro- and anti-inflammatory proteins identified in the literature[12–15], which had been previously developed into an in-house targeted proteomic neuroinflammatory panel. Using this approach, we created a targeted proteomic panel, including biomarkers from current scientific developments and preliminary findings from our own work[16,17]. This targeted proteomic and multiplexed assay included 121 proteins and aimed to validate biomarkers and probe the pathways identified as being perturbed in the discovery phase. Details can be found in Supplementary Table 2 and Fig. 3.

### Demographics-targeted proteomic validation phase (phase I)
For the targeted proteomics analysis, we used plasma samples, independent from the proteomic discovery step, from 99 individuals recently diagnosed with de novo PD (48 men, 50%, mean age 67 years) and 36 healthy controls (HC; 20 men, 57%, mean age 64 years). This was the main cohort, to which we added further samples for validation that consisted of a heterogeneous group of 41 patients with other neurological diseases (OND) (29 men, 71%, mean age 70 years) and 18 patients with vPSG-confirmed iRBD (10 men, 56%, mean age 67 years). Further details can be found in Table 1 and Fig. 3.

### The identification of biomarkers that were significantly and differentially expressed biomarkers between patients with de novo Parkinson's disease and healthy controls- Targeted proteomic validation phase (phase I)
Our targeted proteomic assay was developed for 121 proteins, 32 of which we consistently and reliably detected in plasma. Of these 32 markers, 23 were confirmed as being significantly and differentially expressed between PD and HC. We identified six differentially expressed proteins in the comparison between iRBD patients and HC and between OND and HC (Fig. 3). Both the de novo PD and iRBD groups demonstrated an upregulated expression of the serine protease inhibitors SERPINA3, SERPINF2 and SERPING1, and of the central complement protein C3. Granulin precursor protein was shown to be downregulated in all three patient groups (PD, iRBD and OND) compared to HC. The OND and PD groups had a shared and upregulated expression of the proteins PTGDS, CST3, VCAM1 and PLD3. Detailed information about the diagnoses of the OND group can be found in Table 1, and detailed information about the proteins can be found in Supplementary Table 2. Figure 4 shows the significantly different proteins as Box-scatter plots.

### The biological significance of the differentially expressed proteins- Targeted proteomic validation phase (phase I)
The involvement of the differentially expressed proteins and their impact on biological processes were evaluated using pathway analysis (Ingenuity Pathway Analysis [IPA], Qiagen). The significantly differentially expressed proteins between PD and HC were used as input, with a fold-change set as the expression observation. We considered pathways as significant if they had an enrichment $p$ value <0.05. At least two of the input proteins were included. Three major pathway clusters were identified and consisted of (i) the expression of serine protease

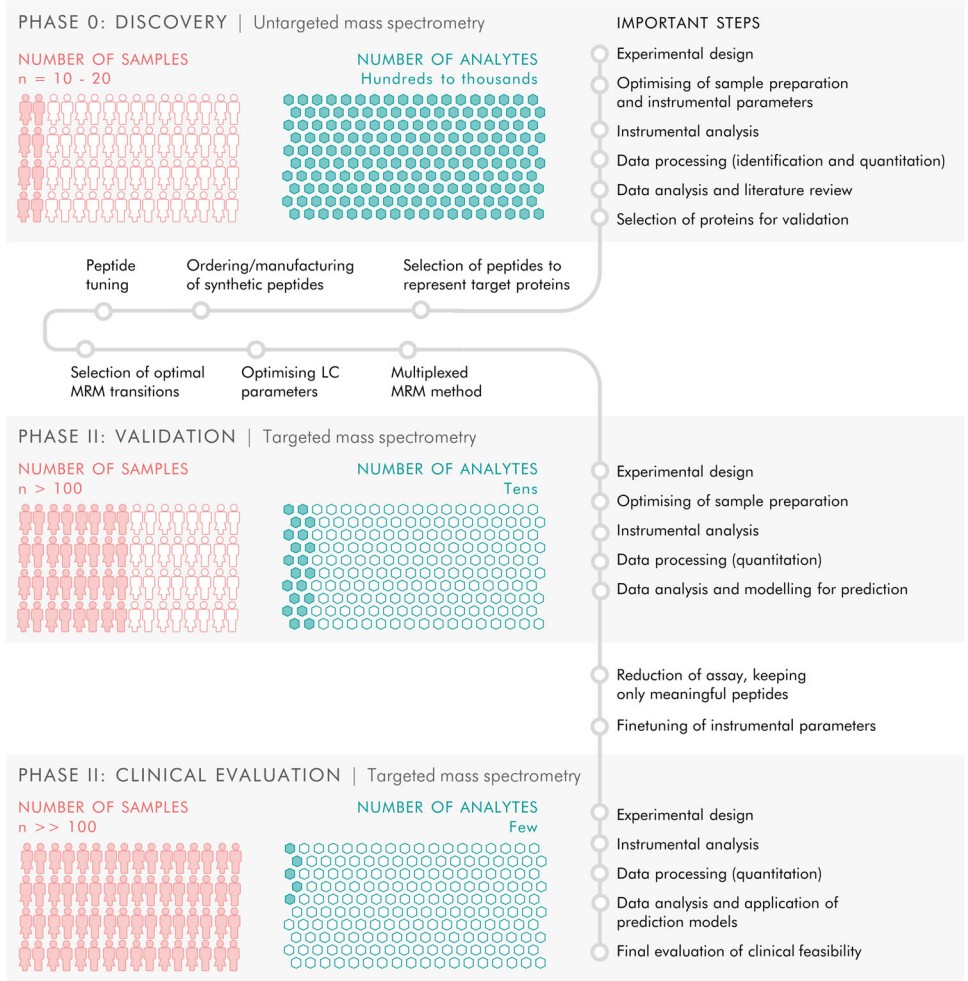

**Fig. 1 | All-over workflow of the study.** The study included three phases. Phase 0 consisted of discovery proteomics by untargeted mass spectrometry to identify putative biomarkers, followed by phase I in which targets from the discovery phase were transferred to a targeted, mass spectrometric MRM method and applied to a new and larger cohort of samples, and finally phase II in which the targeted MRM method was refined and a larger number of samples were analysed to evaluate the clinical feasibility of the targeted protein panel.

inhibitors or serpins and complement and coagulation components, (ii) endoplasmic reticulum (ER) stress/heat shock-related proteins and (iii) the expression of VCAM1, SELE and PPP3CB. The highest enrichment scores were identified in the pathways acute phase response signalling ($p = 7.8$ E$^{-10}$), coagulation system ($p = 7.4$ E$^{-6}$), complement system ($p = 8.1$ E$^{-6}$), LXR/RXR activation ($p = 9.1$ E$^{-6}$), FXR/RXR activation ($p = 9.8$ E$^{-6}$) and glucocorticoid receptor signalling ($p = 2.0$ E$^{-5}$). These are all pathways involved in inflammatory responses. We also identified pathways related to the unfolded protein response ($p = 0.004$) and neuroinflammation ($p = 0.04$), although with lower enrichment scores. For details, see Supplementary Fig. 1.

Inflammation-related pathways (including both the complement system and the acute phase response) demonstrated the highest significance levels, followed by pathways regulating protein folding, ER stress, and heat shock proteins. A network representation of proteins and pathways showed clusters consisting of inflammation/coagulation/lipid metabolism (FXR/RXR and LXR/RXR), heat shock proteins/protein misfolding, and more heterogenous pathway clusters related to Wnt-signalling and extracellular matrix proteins. Figure 5 illustrates the potential detrimental and protective mechanisms suggested to be taking place based on the protein expressions observed in this study, leading to oligomerisation and accumulation of α-synuclein in neuronal Lewy body inclusions and, finally, dopaminergic neuronal cell loss.

## Multivariate analysis shows differences between the proteomes of Parkinson's disease and controls- Targeted proteomic validation phase (phase I)

Principal component analysis (PCA) demonstrated that the HC and PD groups formed two clusters separate from each other over the first and second principal components (PC), attributed with 23.5% and 13.9% of the model's total variance, respectively. The iRBD group was situated in the middle of HC and PD, and the OND group varied considerably with no evident clustering, as expected due to the heterogeneity of diseases. The corresponding loadings of PC1 and PC2 demonstrated that those with PD correlated with lower levels of PPP3CB, DKK3, SELE and GRN, and higher levels of most of the other proteins. The loadings plot had a high level of covariation in the expression of the PPP3CB, DKK3 and SELE proteins, which were all downregulated in PD. These proteins correlated negatively with the expression of SERPINs, complement C3 and HPX, which all showed a high degree of covariation, and were upregulated in the PD group. Data are displayed in Supplementary Fig. 2.

## The use of multiplexed protein panels of protein biomarkers for the prediction of de novo Parkinson's disease- Targeted proteomic validation phase (phase I)

We next applied machine learning to construct a discriminant OPLS-DA model using the PD and HC samples from the validation phase. The

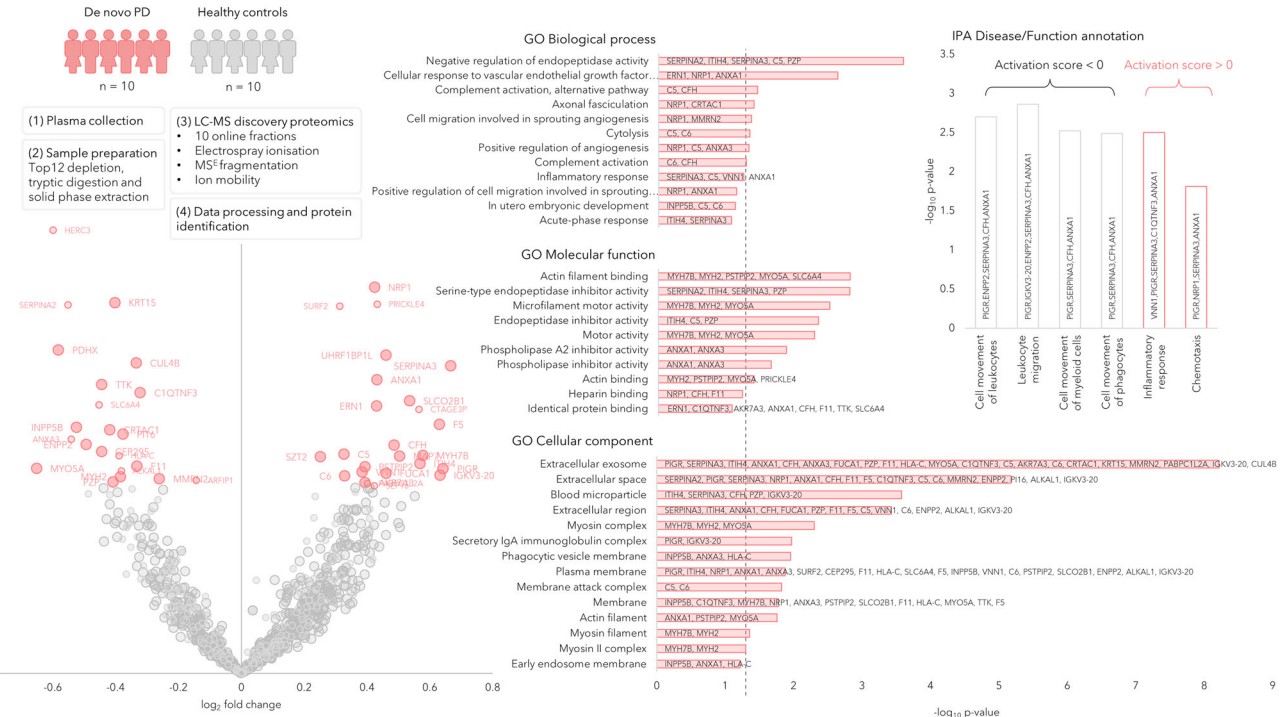

**Fig. 2 | Discovery phase in plasma samples of de novo PD (*n* = 10) and healthy controls (*n* = 10) represented by a Volcano plot showing the protein expression differences between PD and controls (phase 0).** The circle radii in the Volcano plot represent the identification certainty, where large radii represent proteins identified by at least two unique peptides and an identification score >15, smaller radii are given for proteins identified by two or more unique peptides or a confidence score >15. The horizontal axis shows log$_2$ of the average fold-change and the vertical axis shows −log$_{10}$ of the *p* values. The significantly different proteins are annotated by gene name and coloured in pink, while the non-significant proteins are coloured in grey. GO annotations for the significant proteins are shown, the dashed line represents *p* = 0.05. Disease and function annotations from IPA are shown, divided into annotations with a positive or negative activation score. Source data are provided as a Source Data file.

samples clustered into two distinct and well-separated classes, and evaluation of the model showed that it was highly significant ($p = 2.3E^{-27}$ permutations $p = <0.001$). The proteins with the greatest influence on the class separations were GRN, DKK3, C3, SERPINA3, HPX, SERPINF2, CAPN2, SERPING1 and SELE. We predicted the iRBD samples in the model, which resulted in 13 subjects classified as PD (72%) and five not belonging to either group. None of the iRBD samples were classified as controls. We additionally predicted the OND samples, out of which nine were classified as HC, 12 as PD and 19 were not classified as belonging to either group. The 12 samples predicted as PD did not demonstrate enrichment according to the OND groups. The random distribution of the OND samples between PD and HC indicates that the heterogenous group of OND individuals does not share a distinct protein expression with either the HC or PD groups. The iRBD samples that were classified as PD, and not as HC, strongly suggest a shared proteomic profile between iRBD and the protein expression observed in the newly diagnosed PD patients.

We subsequently explored if the observed protein expressions could be used to build a regression model capable of predicting whether individuals belonged to the PD or HC groups. We identified a panel of proteins that discriminated between PD and HC with 100% accuracy and then constructed a linear support vector classification model and applied recursive feature elimination to pinpoint the most discriminating variables. The data were divided into two parts: one consisting of 70% for model training and one containing 30% for testing. The proportion of PD and control samples was maintained in each part. The number of features included in the model was determined by feature ranking with cross-validated recursive feature elimination in the training dataset. The feature selection resulted in a model with eight predictors: GRN, MASP2, HSPA5, PTGDS, ICAM1, C3, DKK3 and SERPING1. The training data

were predicted in the model and resulted in all samples being classified in the correct class. We further constructed receiver operating characteristic (ROC) and precision-recall (PR) curves to illustrate the ability of each protein to distinguish between PD and HC and compared this with the ability of the combined multiplexed protein panel. The combined panel achieved an AUC of 1.0 on both ROC and PR curves. The AUC of the individual predictors ranged from 0.53 to 0.92 in the ROC curve, and from 0.79 to 0.96 in the PR curve (Fig. 6). We further evaluated the whole dataset by performing repeated cross-validation with six splits of the data and 40 repetitions. The resulting classification metrics (Supplementary Fig. 3) demonstrated average and standard deviation for precision, recall, F1 score, and balanced accuracy score of 0.87 ± 0.09, 0.87 ± 0.08, 0.86 ± 0.09 and 0.82 ± 0.12, respectively, thereby indicating a highly robust classification model. Testing the model's specificity for PD, we predicted the heterogenous group of OND, resulting in 26 of the 42 samples being classified as PD-like. Prediction of the prodromal iRBD group resulted in 17 of 18 samples being classified as PD-like. We compared the prediction of the OND and iRBD samples between the OPLS-DA and SVM models, finding that most of the samples were classified in the same group in both models (out of the samples with a classification in the OPLS-DA model: 82% in OND and 100% in iRBD). The proportion of iRBD samples classified as PD in our models (72% in the OPLS-DA model and 94% in the SVM model) is in line with clinical evidence based on longitudinal cohort studies, reporting that over 80% of iRBD subjects will develop an advanced NSD with motor impairment and/or cognitive decline[18]. We evaluated the influence of age and sex on the proteins included in the support vector model and found that neither influenced the model's classification ability (see Supplementary Methods 2 for details).

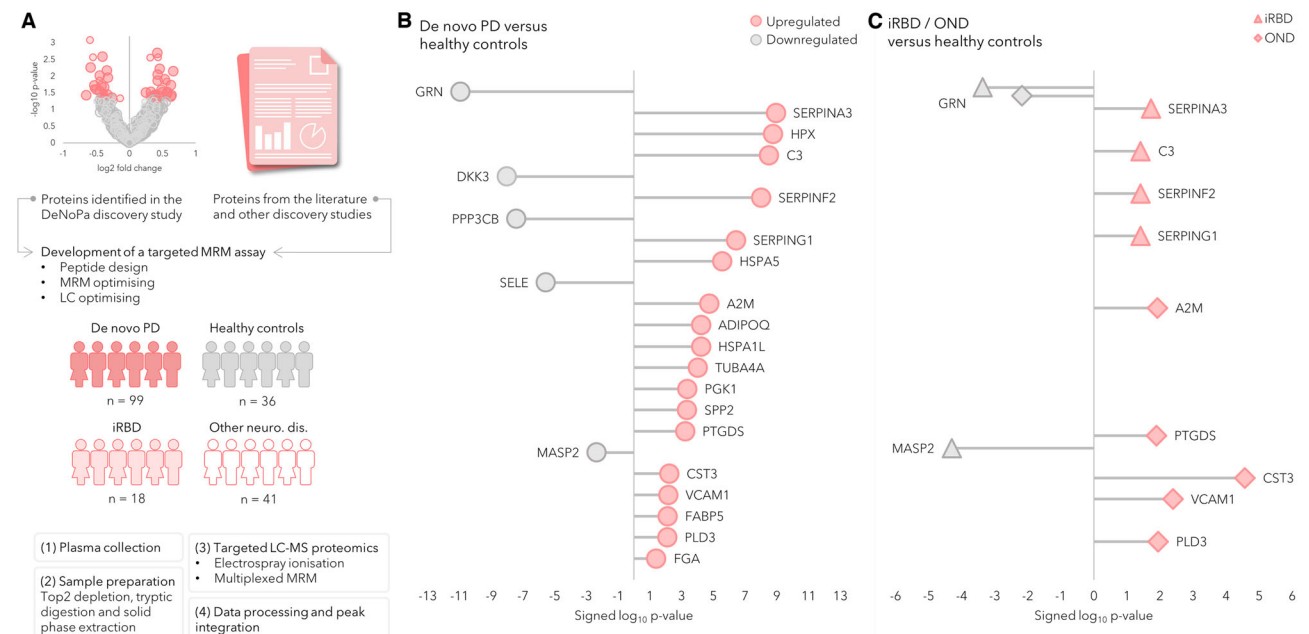

**Fig. 3 | Workflow and overview of results of the targeted proteomic analysis.** Workflow and overview of the results of the targeted proteomic analysis of de novo Parkinson's disease (PD) subjects, healthy controls (HC), and the validation cohorts of other neurological disorders (OND) and isolated REM sleep behaviour disorder (iRBD). **A** A targeted mass spectrometric proteomic assay was developed and optimised. The assay was then applied to plasma samples from cohorts comprising de novo PD (n = 99) and HC (n = 36), and validated in patients with OND (n = 41) and prodromal subjects with iRBD (n = 18). The protein expression difference between

the groups was compared using Mann–Whitney's two-sided U-test with Benjamini–Hochberg FDR adjustment at 5%. The lollipop charts show the $\log_{10} p$ values, signed according to fold-changes. Pink icons represent a protein upregulated in an affected group and grey represents a protein upregulated in controls. **B** Significantly differentially expressed proteins in the comparison between de novo PD and healthy controls. **C** Significantly differentially expressed proteins between iRBD, OND and HC. Source data are provided as a Source Data file.

## Table 1 | Demographics of the samples analysed in the targeted assay (phase I)

| | De novo PD | Healthy controls | Other neurological disorders (OND) | REM sleep behaviour disorder (iRBD) | p value (compared to healthy controls) | | |
| --- | --- | --- | --- | --- | --- | --- | --- |
| | | | | | De novo PD | iRBD | OND |
| Number of subjects | 99 | 36 | 41 | 18 | | | |
| Sex (M/F) | 49/50 | 20/15 | 29/12 | 10/8 | | | |
| M/F% | 49/51 | 57/43 | 71/29 | 56/44 | | | |
| Age (mean ± SD) | 67.1 ± 10.6 | 63.7 ± 6.5 | 70 ± 8.9 | 67.3 ± 8.3 | 8.3E-2 | 8.9E-2 | 8.4E-4 |
| Range | 41–87 | 52–77 | 49–82 | 51–77 | | | |
| UPDRS I (mean ± SD) | 1.8 ± 1.8 | 0.5 ± 0.9 | N/A | 2.9 ± 2.5 | 6.1E-5 | 4.6E-6 | N/A |
| Range | 0–8 | 0–4 | N/A | 0–9 | | | |
| UPDRS II (mean ± SD) | 8.7 ± 6.6 | 0.03 ± 0.2 | N/A | 3.1 ± 3.7 | 1.9E-12 | 1.4E-5 | N/A |
| Range | 0–29 | 0 – 1 | N/A | 0–11 | | | |
| UPDRS III (mean ± SD) | 24.8 ± 13.9 | 0.3 ± 0.7 | 21.2 ± 13.3 | 2.6 ± 2.7 | 7.1E-19 | 1.6E-5 | 1.7E-13 |
| Range | 5–69 | 0–3 | 3–51 | 0–10 | | | |
| UPDRS total score (mean ± SD) | 35.2 ± 18.3 | 0.7 ± 1.3 | N/A | 8.5 ± 7.2 | 1.4E-20 | 8.9E-8 | N/A |
| Range | 6–86 | 0–6 | N/A | 1–26 | | | |
| MMSE total score (mean ± SD) | 28 ± 2.2 | 28.9 ± 1.4 | 26.6 ± 2.6 | 28.5 ± 1.8 | 2.8E-2 | 3.6E-1 | 2.5E-5 |
| Range | 25–30 | 26–30 | 19–30 | 25–30 | | | |

OND consists of subjects with vascular parkinsonism (n = 10), essential tremor (n = 7), progressive supranuclear palsy; PSP (n = 7), multiple system atrophy; MSA (n = 3), corticobasal syndrome; CBS (n = 2), dementia with Lewy bodies; DLB (n = 2), drug-induced tremor (n = 2), dystonic tremor (n = 2), restless legs syndrome (n = 1), hemifacial spasm (n = 1), motoneuron disease (n = 1), amyotrophic shoulder neuralgia (n = 1), Alzheimer's disease (n = 1). The significance between controls and the disease groups was tested by applying Student's two-tailed t-test.
*MMSE* mini-mental state examination, *UPDRS* unified Parkinson's disease rating scale.

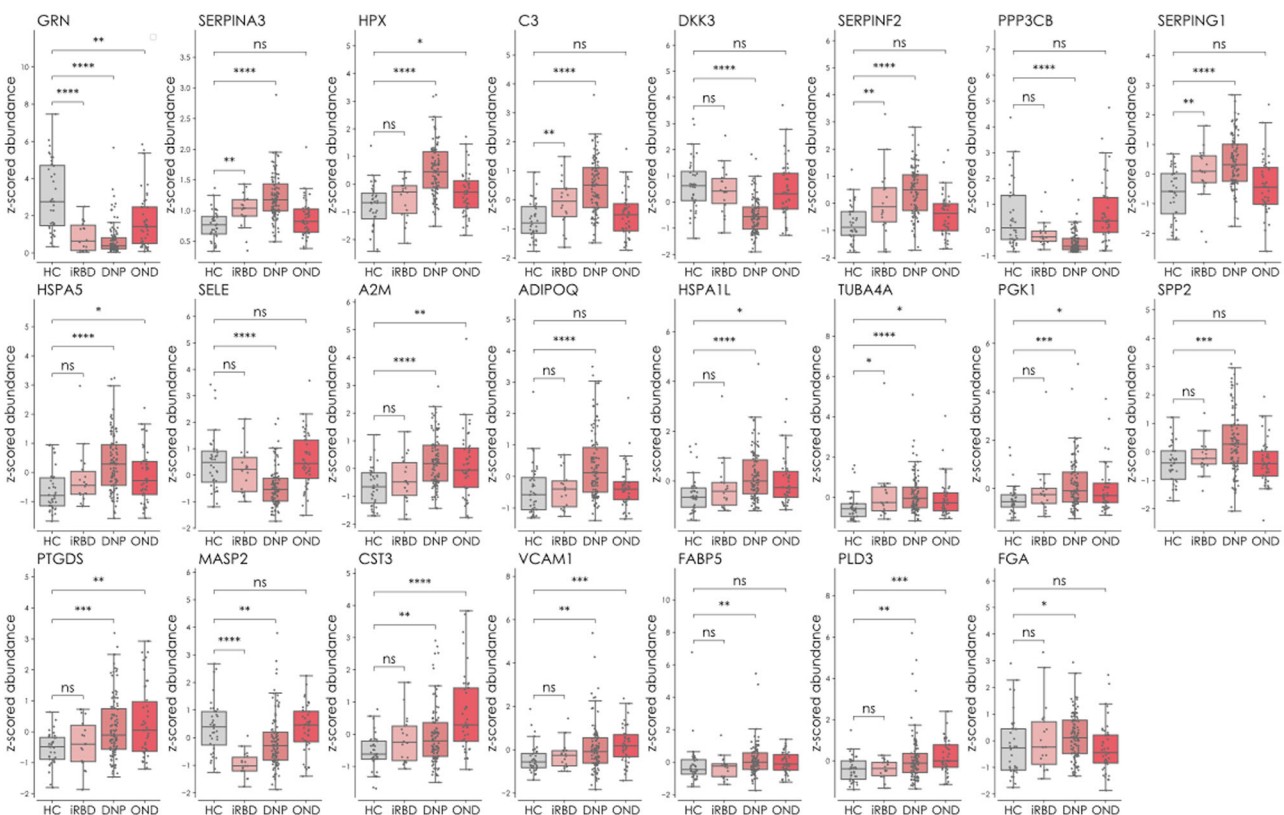

**Fig. 4 | Significantly different proteins between controls and the different disease groups de novo PD (DNP), iRBD and OND (phase II).** The data are displayed as Box and Whisker plots overlaid with scatter plots of the individual measurements. The whiskers show the minimum and maximum, and the boxes show the 25th percentile, the median and the 75th percentile. The protein expression difference between the groups was compared using Mann–Whitney's two-sided $U$-test with Benjamini–Hochberg multiple testing correction (FDR adjustment at 5%). ns not significant, $*p < 0.05$, $**p < 0.01$, $***p < 0.001$ and $****p < 0.0001$. The proteins are represented by gene names. Source data are provided as a Source Data file.

## Development of a rapid and refined LC-MS/MS method and evaluation of an independent and longitudinal iRBD cohort (Independent replication cohort-phase II)

To evaluate the results from the initial prediction models focusing on at-risk subjects, we developed and refined our targeted and multiplexed proteomic test to quantitate only those proteins that were readily and reliably detectable from the initial targeted proteomic assay ($n = 32$). Next, we analysed an additional set of 146 longitudinal samples from an independent cohort of 54 individuals with iRBD. This cohort was available from continuing recruitment at the same centre and consisted of longitudinally followed iRBD subjects. Deep phenotyping revealed 100% (54/54) had RBD on PSG, 88.9% (48/54) had hyposmia as identified with the Sniffin' Stick Identification Test, and 91.7 % (22/24) had neuronal α-synuclein positivity as shown by α-synuclein Seed Amplification Assay (SAA) in cerebrospinal fluid (CSF)[19]. Longitudinal follow-up was available for up to 10 years, during which 16 subjects (20%) phenoconverted to either PD ($n = 11$) or dementia with Lewy bodies (DLB; $n = 5$). Since only serum samples were available from the independent replication cohort (further details can be found in Supplementary Table 3), we investigated how the proteins in our assay correlated between plasma, serum, and CSF and found good correlations between plasma and serum, but poor correlations between these blood matrices and CSF. The limited correlations between blood and CSF proteins correspond to those of other studies comparing the protein expression between plasma/serum and CSF[20,21] and underscore that our test does not necessarily reflect a prodromal and PD-specific proteomic signature of the protein expression in the CSF in proximity to the brain, but rather shows an earlier change in the blood protein expression between healthy status and very early PD

patients (Details from this comparison can be found in Supplementary Methods 1 and Supplementary Fig. 4).

We applied all available longitudinal iRBD samples ($n = 146$) from phase II to the two machine-learning models (OPLS-DA and support vector machine) constructed in phase I (PD vs. HC). The OPLS-DA model, based on all 32 detected proteins, identified 70% of the iRBD samples as PD, while the SVM model, which was based on a panel of eight proteins, identified 79% of the samples as PD. As mentioned above, at the time of analysis, 16 of the 54 subjects in our longitudinal iRBD validation cohort had developed PD/DLB. The earliest correct classification was 7.3 years prior to phenoconversion and the latest was 0.9 years prior to diagnosis (average $3.5 \pm 2.4$ years). Detailed information can be found in Fig. 7 and Supplementary Methods 3.

## The correlation between differentially expressed protein biomarkers and patients' clinical data in the targeted proteomic validation phase (phase I)

We next evaluated the relationship between proteins and clinical data by correlating the protein expression in PD and HC (from phase I) with clinical scores (Mini-Mental State Examination [MMSE], Hoehn & Yahr stage [H&Y] and UPDRS [Unified Parkinson's Disease Rating Scale; I–III, and total score]). We found negative correlations for GRN, DKK3, PPP3CB, and SELE with H&Y and UPDRS parts II, III, and total score, possibly indicating a connection between a more severe clinical (especially motor) impairment and lower expression of markers in the Wnt-signalling pathways (DKK3 and PPP3CB). Higher Cystatin C plasma levels correlated with higher numbers in UPDRS part III (motor performance) and UPDRS total score. The same was found for PTGDS plasma levels, which were also negatively correlated with MMSE. The

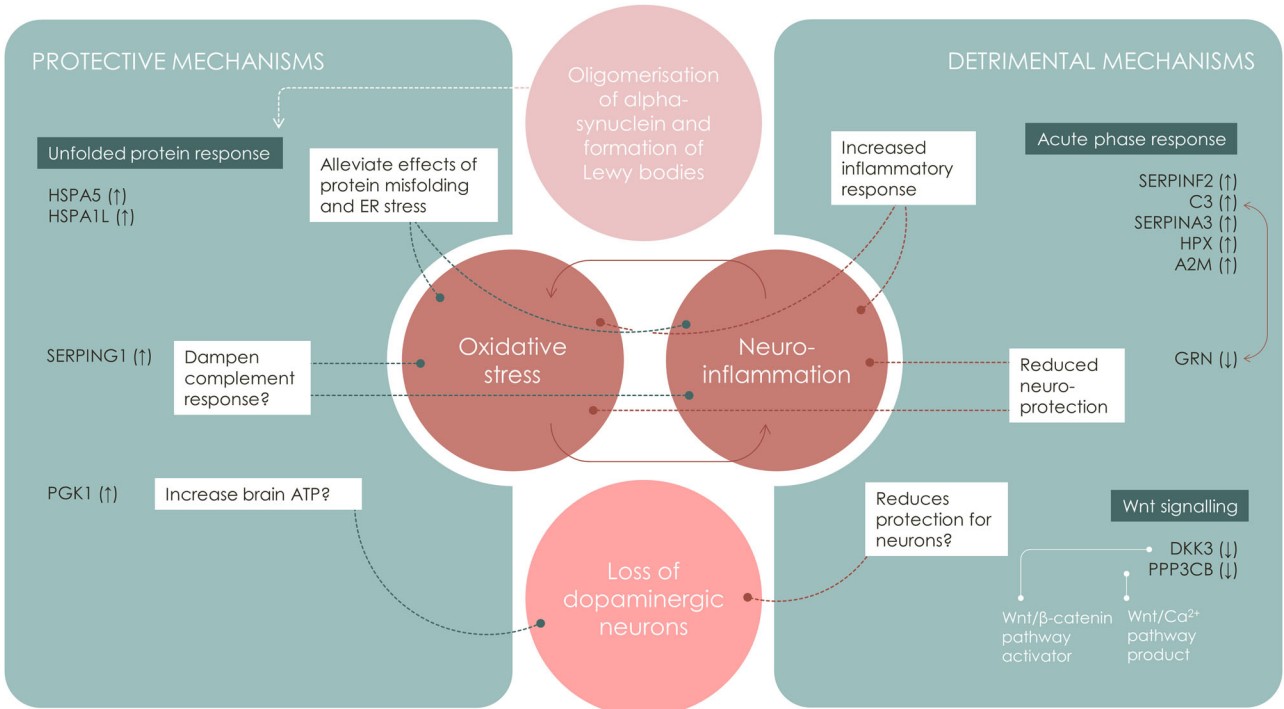

**Fig. 5 | Suggested involvement of the differentially expressed proteins in neuronal synuclein disease.** Oligomerisation and accumulation of α-synuclein in Lewy body inclusions is a key process in the pathophysiology of neuronal synuclein disease, i.e. Parkinson's disease and dementia with Lewy bodies from aggregation and accumulation, the pathological pathway includes different steps finally leading to the loss of dopaminergic neurons. Protective and detrimental mechanisms influence these processes, based on the differently expressed protein profiles, assessed by targeted mass spectrometry in our study. Detailed information about the proteins can be found in Supplementary Table 2.

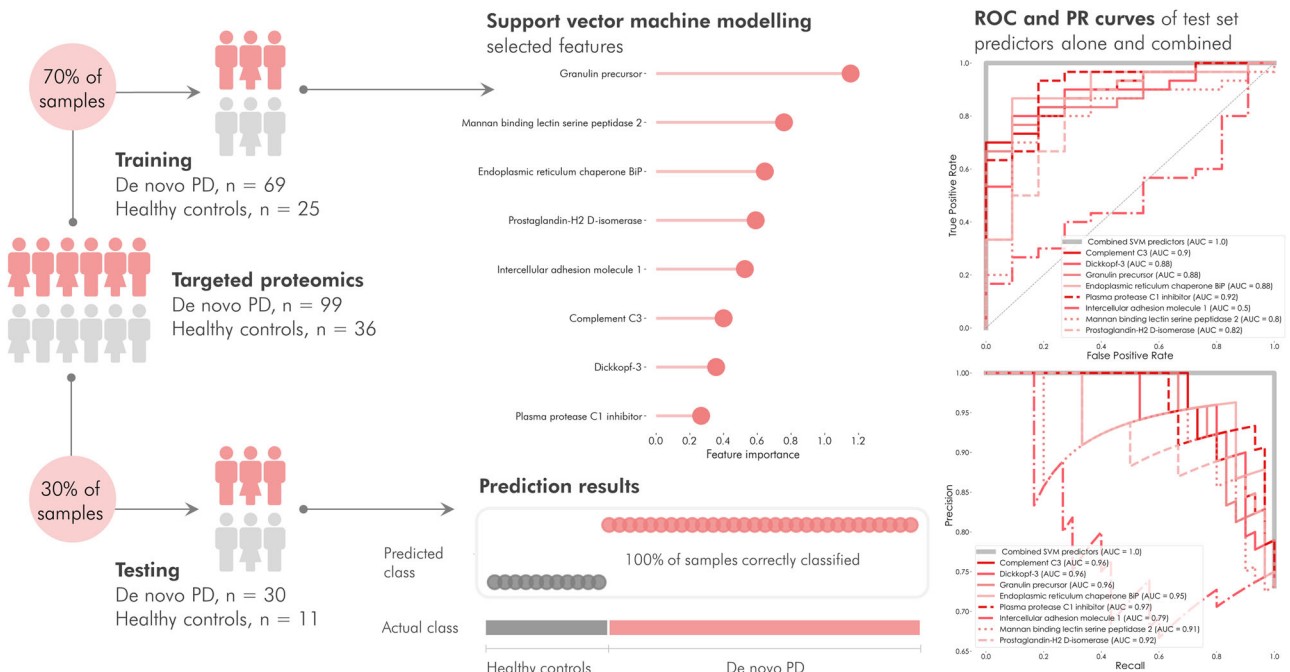

**Fig. 6 | Linear support vector classification of PD and control subjects. (phase I).** The model was trained on 70% of the samples to establish the most discriminating features. Applying cross-validated recursive feature elimination, the top predictors were determined as a granulin precursor, mannan-binding lectin-serine peptidase 2, endoplasmic reticulum chaperone-BiP, prostaglandin-H2 D-isomerase, intercellular adhesion molecule-1, complement C3, dickkopf-3 and plasma protease C1 inhibitor. The remaining 30% of samples were predicted in the model and resulted in 100% prediction accuracy. Receiver operating characteristics (ROC) and precision-recall (PR) curves of the individual and combined proteins in the test set demonstrated that the individual proteins achieved ROC area under the curve (AUC) values 0.53–0.92 and PR values 0.79–0.96, while the combined predictors reached an area under the curve = 1.0. Source data are provided as a Source Data file.

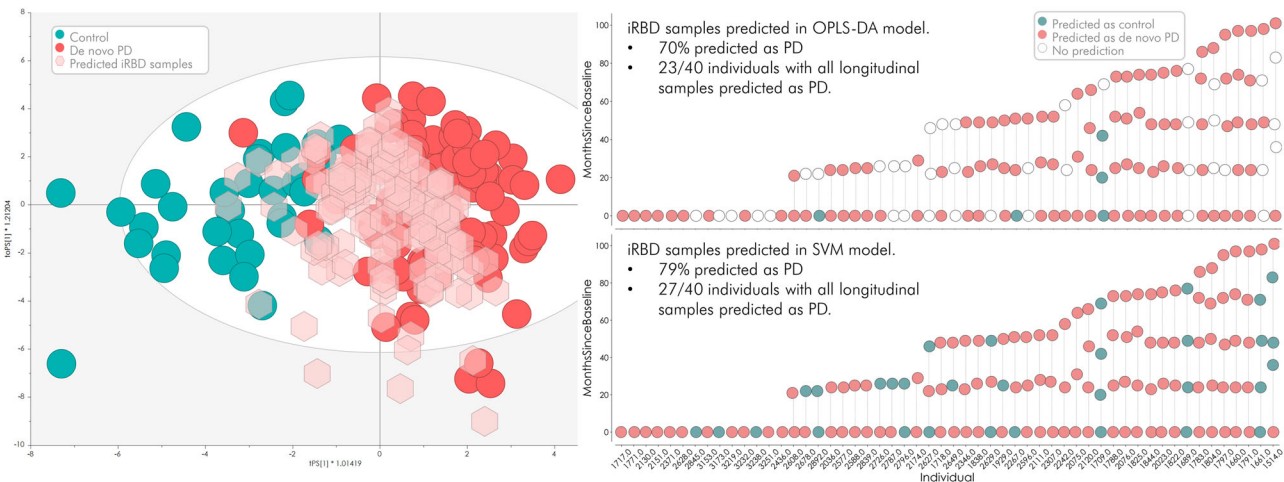

**Fig. 7 | Prediction results from of a newly acquired set of prodromal isolated REM sleep behaviour disorder (iRBD) samples (phase II).** 146 new serum samples from individuals diagnosed with iRBD, several with longitudinal follow-up samples, were predicted in the OPLS-DA model. 70% of the samples were predicted as Parkinson's disease (PD), and 23 of 40 individuals had all their longitudinal samples predicted as PD. In the more refined support vector machine (SVM) model, 79% of the 146 new samples were predicted as PD and 27 of 40 individuals consistently had all their longitudinal samples predicted as PD. Source data are provided as a Source Data file.

central complement cascade protein, C3, negatively correlated with MMSE, and positively correlated with H&Y, UPDRS part III, and total score. The UPR-regulating protein BiP (HSPA5) correlated negatively with MMSE, and positively with H&Y and UPDRS parts II, III, and total score. The ERAD-associated proteins, HSPAIL and adiponectin, were positively correlated with H&Y, and UPDRS parts II, III, and total score. SERPINs (SERPINA3, SERPINF2 and SERPING1) and hemopexin (HPX) correlated negatively with MMSE and positively with H&Y and UPDRS parts II, III, and total score. In general, the MMSE score was inversely correlated with H&Y stage and UPDRS scores. For detailed information, see Fig. 8 and Table 2.

### Comparison of clinical outcomes and measurements in the longitudinal iRBD cohort-Independent replication cohort-phase II

The longitudinal expression in the iRBD samples was evaluated using linear mixed-effects models. Conditional growth models with random slopes and random intercepts between the individuals were constructed. After adjusting the $p$ values for multiple testing by applying the Benjamini−Hochberg (BH) procedure with alpha = 0.05, we found that Butyrylcholinesterase (BCHE) was significantly decreased over the timepoints in the iRBD individuals ($p = 0.01$). We next focused only on the iRBD samples with at least two timepoints and for which PD had consistently been predicted in the SVM model ($n = 90$). This produced comparable results to the initial model with BCHE significantly related with time since baseline ($p = 0.01$), but also TUBA4A was nominally significantly increased ($p = 0.04$) although not passing the BH FDR threshold. The modelling also demonstrated that the clinical measurements H&Y ($p = 0.02$), UPDRS I–III ($p = 0.02$), and UPDRS I and III ($p = 0.03$ and 0.03, respectively), were significantly related to the time since baseline in the iRBD group post multiple testing correction. PD non-motor symptoms, as measured on the PD NMS sum score, were strongly correlated with longitudinal motor progression ($p = 5E^{-8}$). Similarly, the questionnaire for quality of life PDQ-39's mean values also correlated with longitudinal motor progression ($p = 0.005$). From available routine blood values, cholesterol was associated with longitudinal timepoints ($p = 0.02$). Details can be found in Supplementary Table 4. Correlating the clinical measurements with the targeted proteomic data, we applied Spearman's correlation and found that cholesterol was positively correlated with six of the identified proteins (Supplementary Table 5), including HSPA8, APOE and MASP2 ($p = 5E^{-9}$,

0.0003 and 0.003, respectively). Also significantly correlated, but to a lesser degree and not passing the BH FDR threshold, were the PD NMS sum which correlated negatively with TUBA4A ($p$ unadjusted = 0.01) and the PDQ-39 mean values, which correlated negatively with CST3 and PTGDS ($p$ unadjusted = 0.03 and 0.05, respectively).

## Discussion

PD has emerged as the world's fastest-growing neurodegenerative disorder and currently affects close to 10 million people worldwide. Consequently, there is an urgent need for disease-modifying and prevention strategies[22,23]. The development of such strategies is hampered by two limitations: there are major gaps in our understanding of the earliest events in the molecular pathophysiology of PD, and we lack reliable and objective biomarkers and tests in easily accessible biofluids. We, therefore, need biomarkers that can identify PD earlier, preferably a significant time before an individual develops significant neuronal loss and disabling motor and/or cognitive disease. Such biomarkers would advance population-based screenings to identify individuals at risk and who could be included in upcoming prevention trials.

In the last years, CSF SAA emerged as the most specific indicator for NSD, in prodromal stages like iRBD, with an impressively high sensitivity and specificity of up to 74 and 93%, respectively, across various cohorts[9,24]. Despite the many questions surrounding SAA that need to be answered, including the ultimate understanding of its functionality, it is a true milestone for advancing prevention trials. It is, however, hampered by having only been shown to be robust in CSF and by the slow development and high variability of SAA in peripheral blood[25], as well as by the lack of quantification capabilities. An easier and more accessible biofluid test would enable screening large population-based cohorts for at-risk status to develop an NSD. Therefore, the identification of additional biomarkers is needed, as is further knowledge of the biomarkers and pathways of the underlying pathophysiology (e.g. inflammation) during the earliest stage of NSD.

Other emerging multiplex technologies are increasingly used to identify individual proteomic biomarkers. However, these techniques are not true proteomic or 'eyes open' methods, as they rely on selected large panels of specific antibodies/and other (e.g. aptamer)-based assay technologies. These techniques, although useful, have not provided consistent results[3,26]. Proteomics using mass spectrometry measures all expressed proteins in an unbiased fashion as opposed to

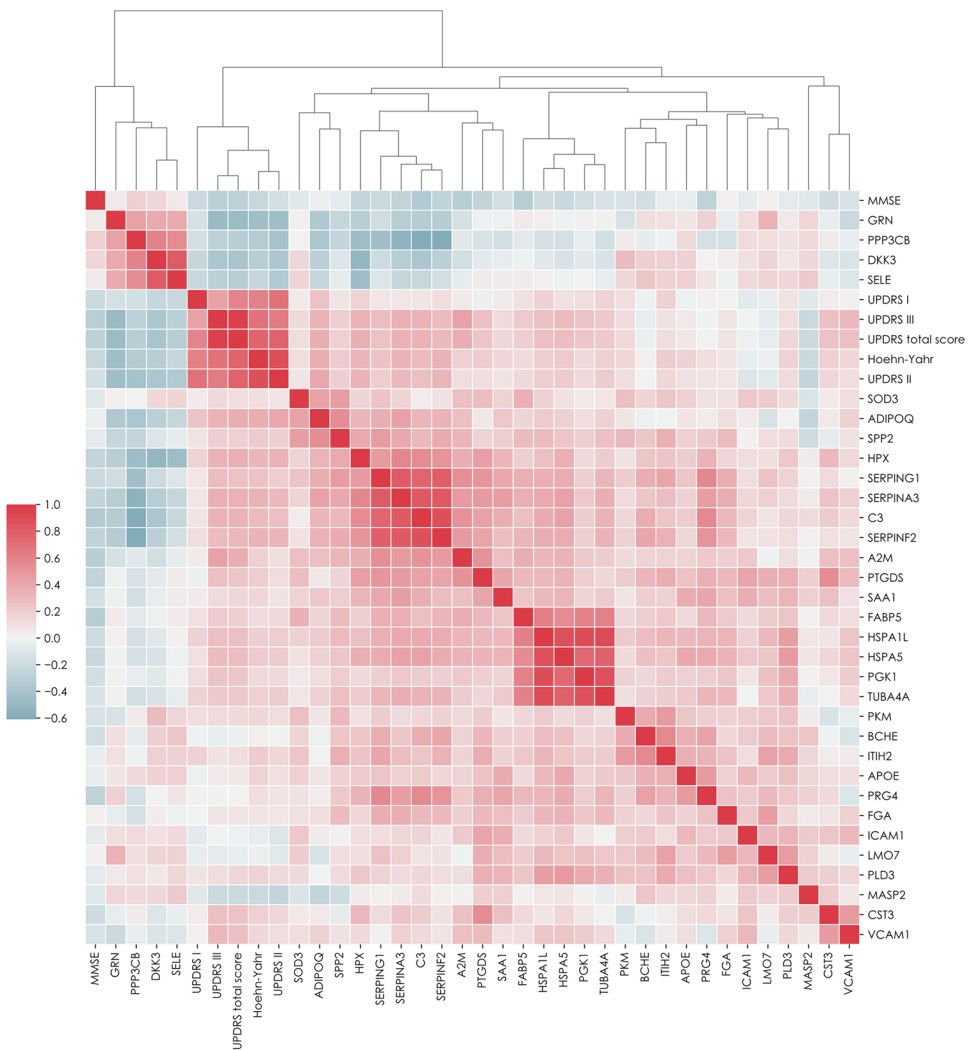

**Fig. 8 | Correlation and clustering heatmap of proteins measured by targeted mass spectrometry and clinical scores in controls and Parkinson's disease subjects. (phase I).** The correlation was performed using Spearman's procedure, and the clustering method was set to average. The clustering metric was Euclidean. The heatmap is coloured by correlation coefficient where red represents positive and blue negative correlations. The proteins are represented by gene names. Detailed information about the protein correlations can be found in Supplementary Table 3. De novo Parkinson's disease ($n = 99$) and healthy controls ($n = 36$). MMSE mini-mental state examination, UPDRS unified Parkinson's disease rating Scale. Source data are provided as a Source Data file.

those selectively included in a panel that also includes variability due to cross-reactivity. Therefore, proteomic screening using mass spectrometry-based techniques is much more likely to identify pathways or biomarkers and provides more meaningful insights into the disease mechanisms involved in PD. We found a discrepancy between the detected markers during the discovery and the targeted phases. This is a known phenomenon in biomarker translation[27] that is also reflected in the low number of biomarkers having received FDA approval[28]. We addressed this by using previously reported successful improvement strategies in proteomic approaches, namely by refining our panel, reducing the number of markers, and increasing the sample size[29]. Furthermore, the validation of potential biomarkers was performed on a second and different type of mass spectrometer (triple quadrupole), which has the advantage of being available in all large hospitals.

Targeted MS has been previously applied in PD, including by the current authors, but the biological fluid used in the majority of studies is CSF[30] and not peripheral fluids such as blood. Here we demonstrate that even with a very low required volume of plasma/serum (10 µl) targeted proteomic is feasible.

The targeted proteomic assay presented here was developed from proteins identified in an unbiased discovery study, from our previous research, and from the literature. It included several inflammatory markers, Wnt-signalling members, and proteins indicative of protein misfolding. When analysing PD, OND, iRBD and HC in the targeted proteomic validation phase, we identified and confirmed 23 distinct and differentially expressed proteins between PD and HC. Our analysis moreover demonstrated that iRBD possesses a significantly different protein profile compared to HC, consisting of decreased levels of GRN and MASP2 and increased levels of the complement factor C3 and SERPINs (SERPINA3, SERPINF2 and SERPING1), thus indicating early involvement of inflammatory pathways in the initial pathophysiological steps of PD. Comparing these results to previous findings by our and other groups[8,31] highlights the link between these proteins and the pathways of complement activation, coagulation cascades, and Wnt-signalling.

By applying machine-learning models, we classified and separated de novo PD or control samples with 100% accuracy based on the expression of eight proteins (GRN, MASP2, HSPA5, PTGDS, ICAM1, C3, DKK3 and SERPING1).

**Table 2 | $p$ values from the Spearman correlation of proteins measured by targeted mass spectrometry and clinical PD assessment scores in PD subjects and healthy controls (phase I)**

| | MMSE | | Hoehn-Yahr | | UPDRS I | | UPDRS II | | UPDRS III | | UPDRS total score | |
|---|---|---|---|---|---|---|---|---|---|---|---|---|
| MMSE | | | 4.0E-2 | (−0.23) | | | | | 3.0E-3 | (−0.31) | 5.3E-3 | (−0.29) |
| Hoehn-Yahr | | | | | 1.0E-13 | (−0.61) | 1.3E-42 | (−0.88) | 8.1E-19 | (−0.69) | 7.1E-26 | (−0.77) |
| UPDRS I | | | 5.2E-14 | (−0.61) | | | 2.7E-19 | (−0.7) | 8.1E-7 | (−0.44) | 7.6E-13 | (−0.59) |
| UPDRS II | | | 1.3E-42 | (−0.88) | 8.2E-19 | (−0.7) | | | 1.3E-15 | (−0.64) | 9.6E-27 | (−0.78) |
| UPDRS III | 1.6E-2 | (−0.31) | 5.4E-19 | (−0.69) | 1.1E-6 | (−0.44) | 9.6E-16 | (−0.64) | | | 2.0E-77 | (−0.97) |
| UPDRS total score | 2.1E-2 | (−0.29) | 1.1E-25 | (−0.77) | 1.0E-12 | (−0.59) | 9.6E-27 | (−0.78) | 2.0E-77 | (−0.97) | | |
| A2M | 1.2E-2 | (−0.32) | | | | | | | 8.1E-7 | (−0.44) | 6.7E-5 | (−0.37) |
| ADIPOQ | | | 6.8E-5 | (−0.37) | 2.4E-2 | (−0.25) | 9.4E-6 | (−0.4) | 2.0E-4 | (−0.34) | 5.5E-5 | (−0.37) |
| C3 | 9.8E-3 | (−0.35) | 1.7E-3 | (−0.3) | | | 3.2E-3 | (−0.28) | 2.0E-4 | (−0.34) | 2.6E-4 | (−0.34) |
| CST3 | | | | | | | | | 2.6E-3 | (−0.28) | 5.0E-3 | (−0.26) |
| DKK3 | | | 5.3E-4 | (−0.32) | 3.8E-2 | (−0.24) | 3.2E-5 | (−0.38) | 2.8E-5 | (−0.38) | 1.5E-5 | (−0.4) |
| FABP5 | 1.2E-2 | (−0.32) | | | | | | | 1.9E-2 | (−0.22) | 4.0E-2 | (−0.19) |
| GRN | | | 4.8E-7 | (−0.45) | | | 3.8E-7 | (−0.45) | 2.8E-8 | (−0.49) | 3.3E-8 | (−0.48) |
| HPX | 4.6E-2 | (−0.25) | 5.3E-4 | (−0.32) | | | 2.2E-4 | (−0.34) | 2.1E-4 | (−0.34) | 9.2E-5 | (−0.36) |
| HSPA1L | | | 4.5E-2 | (−0.19) | | | 4.6E-2 | (−0.2) | 2.3E-3 | (−0.28) | 2.7E-3 | (−0.28) |
| HSPA5 | | | 3.0E-2 | (−0.21) | | | 4.6E-2 | (−0.2) | 1.2E-3 | (−0.3) | 2.2E-3 | (−0.29) |
| ITIH2 | | | 3.2E-2 | (−0.21) | | | | | | | | |
| MASP2 | | | 3.0E-2 | (−0.21) | | | 1.1E-2 | (−0.25) | 1.9E-2 | (−0.22) | 3.1E-2 | (−0.2) |
| PGK1 | | | | | | | | | 5.6E-3 | (−0.25) | 1.0E-2 | (−0.24) |
| PPP3CB | | | 3.4E-4 | (−0.34) | | | 3.1E-6 | (−0.42) | 2.6E-3 | (−0.28) | 5.3E-4 | (−0.32) |
| PRG4 | 3.1E-2 | (−0.27) | | | | | | | | | | |
| PTGDS | 3.2E-2 | (−0.26) | 4.0E-2 | (−0.2) | | | | | 2.6E-3 | (−0.28) | 8.3E-3 | (−0.24) |
| SELE | | | 1.3E-2 | (−0.24) | 4.7E-2 | (−0.23) | 1.3E-4 | (−0.35) | 1.4E-3 | (−0.3) | 5.1E-4 | (−0.32) |
| SERPINA3 | 3.2E-2 | (−0.26) | 3.8E-4 | (−0.33) | | | 7.9E-4 | (−0.31) | 1.5E-4 | (−0.35) | 1.4E-4 | (−0.35) |
| SERPINF2 | 3.1E-2 | (−0.27) | 3.4E-4 | (−0.34) | | | 3.8E-4 | (−0.33) | 1.2E-3 | (−0.3) | 1.1E-3 | (−0.3) |
| SERPING1 | | | 7.5E-03 | (−0.26) | | | 4.6E-02 | (−0.2) | 2.0E-3 | (−0.29) | 4.6E-3 | (−0.26) |
| SPP2 | | | 3.0E-2 | (−0.21) | | | | | 3.7E-2 | (−0.19) | 4.7E-2 | (−0.19) |
| TUBA4A | | | | | | | 4.6E-2 | (−0.2) | 1.7E-2 | (−0.22) | 1.6E-2 | (−0.22) |
| VCAM1 | | | | | | | | | 8.0E-4 | (−0.31) | 3.0E-3 | (−0.28) |

The correlation significance was determined by the Student's two-tailed $t$-test. The $p$ values were adjusted for multiple comparisons using the Benjamini–Hochberg procedure with alpha = 0.05. The $p$ values are followed by correlation coefficients in parenthesis.
*MMSE* mini-mental state examination, *UPDRS* unified Parkinson's disease rating scale.

With an independent validation, we added (a) a larger sample set and (b) longitudinal samples from the most interesting subgroup with 54 iRBD subjects and a total of 146 serum samples. We were able to validate our previous panel with a high prediction rate (79%) of these individuals as seen in PD in the targeted approach. Interestingly, the biomarker panel itself did not correlate with longitudinal expression but remained robust after the initial classification of iRBD. So far, 16 of the 54 iRBD subjects converted to PD/DLB (stage 3 NSD). Out of these samples, the SVM model predicted ten individuals with all their time-points classified as PD, and of the 11 iRBD subjects who converted to PD/DLB, eight were identified as PD by the proteome analysis. Our panel, therefore, identified a PD-specific change in blood up to 7 years before the development of the stage 3 NSD.

The main shortcoming with many previously explored PD biomarkers is weak or no correlation with clinical progression data. So far, outcome measures in clinical trials are primarily based on motor progression, often by a clinical rating scale such as the UPDRS and/or wearable technologies. More objective biomarkers correlating with or reflecting the progression of the pathophysiology and clinical symptoms would be of the utmost importance. We, therefore, calculated correlations with clinical parameters and identified an association with multiple markers, including DKK3, PPP3CB and C3, indicating downregulation of Wnt-signalling pathways. Increased activity of the complement cascade correlated with higher scores in symptom severity (UPDRS part III and total score) and lower scores in cognitive performance (MMSE).

Protein (i.e. α-synuclein) misfolding is a well-known component of PD pathology and is believed to be the key factor behind Lewy body formation[32]. The transport of excessive amounts of misfolded proteins or increased folding cycles can induce ER stress. A cellular defence mechanism to alleviate ER stress is the unfolded protein response (UPR) reducing ER protein influx and increasing protein folding capacity[33]. The UPR is mainly activated by BiP-bound misfolded proteins[34]. The higher expressed markers HSPA5 (UPR-regulating protein BiP) and HSPA1L in our plasma samples of early PD indicate ER stress as a significant factor in the disease process and has been previously linked to PD in both mouse models and brain tissue studies[35,36].

As mentioned by other groups and confirmed in our results, increasing evidence suggests inflammation is a specific feature in early PD. Complement activation has been associated with the formation of α-synuclein and Lewy bodies in PD and deposits of the complement factors iC3b and C9 have been found in Lewy bodies[37]. C3 is a central molecule in the complement cascade and was highly upregulated in blood in both PD and both independent iRBD sample sets analysed in this study. This upregulation in the earliest phase of motor PD (stage 3 NSD), and even in the prodromal phase (stage 2 NSD), clearly indicates

inflammation as an early, if not the initial, event in PD neurodegeneration. Complement C3 levels correlated positively with indicators of motor dysfunction (H&Y stage and UPDRS)—indicating a direct connection between high plasma levels of inflammatory proteins and motor symptoms—and negatively with cognitive decline, here with the MMSE.

The protein Mannan-binding serine peptidase 2 (MASP2), an initiator of the lectin part of the complement cascade, was significantly downregulated in PD and iRBD. MASP1 and MASP2 proteins are inhibited by plasma protease C1 inhibitor SERPING1 in the lectin pathway, with SERPING1 modulating the complement cascade as it belongs to the SERPIN family of acute phase proteins[38]. In experimental PD mice models, increased SERPING1 levels are associated with dopaminergic cell death[39]. Acting as a serine/cysteine proteinase inhibitor, SERPING1 can increase serine levels, which could also affect αSyn phosphorylation. This can play a crucial role in PD pathology, as almost 90% of αSyn in Lewy bodies is phosphorylated on Serine129[40,41]. We identified increased SERPING1 plasma levels in both PD and iRBD in our analysis (compared to HC), thus contributing to conditions with increased αSyn phosphorylation, consecutive aggregation, Lewy body formation, and finally degeneration of dopaminergic neurons. Furthermore, we observed a strong correlation of SERPING1 plasma levels with UPDRS II, III and total score, as a direct measure of dopaminergic cell loss[39].

Alpha-2-antiplasmin (SERPINF2) was also significantly upregulated in PD and iRBD. SERPINF2 is a major regulator of the clotting pathway, acting as an inhibitor of plasmin, a serine protease formed upon the proteolytic cleavage of its precursor, plasminogen, by tissue-type plasminogen activator (t-PA) or by the urokinase-type plasminogen activator (u-PA). Plasmin has been reported to cleave and degrade extracellular and aggregated αSyn[42]. Recently, we showed that activation of the plasminogen/plasmin system is decreased in PD, indicated by decreased plasma levels of uPA and its corresponding receptor uPAR, while t-PA was associated with faster disease progression[8]. The upregulation of SERPINF2 observed here is another indicator of decreased plasmin activity. Alpha-1-antichymotrypsin (SERPINA3), a third member of the SERPIN family, was also upregulated in the PD subjects. In the CNS, the primary source of SERPINA3 is astrocytes, where its expression is upregulated by various inflammatory receptor complexes[38].

Overall, independent upregulation of these three members of the SERPIN (SERPING1, SERPINF2, SERPINA3) family is also indicative of increased inflammatory activity, combined with less activation of the plasmin system, and correlation with motor and non-motor symptom severity. In addition, a strong downregulation of progranulin (*GRN*) was detected, indicating a potential loss of neuroprotection and increased susceptibility to neuroinflammation. *GRN* may act as a neurotrophic factor, promoting neuronal survival and modulating lysosomal function. Loss-of-function mutations in the *GRN* gene are a cause of frontotemporal dementia and familial DLB. *GRN* gene variants are also known to increase the risk of developing Alzheimer's disease (AD) and PD[43]. The main characteristics of neurodegeneration related to *GRN* are TDP43(-Transactive response DNA binding protein 43) inclusions, but Lewy body pathology is also very common. Loss of progranulin has further been linked to increased production of pro-inflammatory species such as tumour necrosis factor (TNF) and IL-6 in microglia[15]. A study in mice showed that *Grn-/-* mice had elevated levels of complement proteins, including C3, even before the onset of neurodegeneration[44]. Additionally, previous studies have found GRN downregulated in serum samples of advanced PD compared to AD and healthy individuals[45].

As a possible compensatory reaction to the described increased inflammatory markers, the levels of Prostaglandin-H2 D-isomerase (PTGDS)/Prostaglandin-D2 synthase (PGDS2), better known as β-trace protein, were upregulated. PDGDS is an important brain enzyme

producing prostaglandin D2 (PGD2), which has a neuroprotective and anti-inflammatory function. The upregulation reported here could be a reaction to the amount of neuronal cell loss, which is also seen in the significant correlation with the clinical motor and cognitive scales (see below). Furthermore, β-trace protein is a marker for CSF and is used to identify the fluid in clinical routine diagnostics, thus helping detect CSF leakage[46]. Increased plasma levels could be indicative of a disrupted blood–brain barrier (BBB), often discussed in PD pathology[47] and demonstrated in our cohorts.

Our study shows that the Wnt-related proteins DKK3 and PPP3CB are strongly downregulated in de novo PD. DKK3 is an activator of the canonical Wnt/β-catenin branch and PPP3CB is a component of the non-canonical Wnt/$Ca^{2+}$ signalling pathway. Wnts are secreted, cysteine-rich glycoproteins that act as ligands to locally stimulate receptor-mediated signal transduction of the Wnt-pathway[48]. Wnt-signalling is crucial for the development and maintenance of dopaminergic neurons[49], shows protective effects on midbrain dopaminergic neurons[50], and seems to be involved in the maintenance of the BBB[48,51]. Wnt-ligands and agonists trigger a "Wnt-On" stage, characterised by neuronal plasticity and protection, while the opposite "Wnt-Off" stage, potentially leading to neurodegeneration, triggered by the phosphorylation activity of glycogen synthetase kinase-3β (GSK-3beta)[50,52]. Wnt-inhibitors are separated into secreted Frizzled-related proteins (sFRP) and Dickkopf proteins (DKK). DKK1, DKK2 and DKK4 act as antagonists, while DKK3 is an agonist and activator[53]. Adult neurogenesis is primarily governed by canonical Wnt/β-catenin signaling[54] and downregulation of Wnt-signalling promotes dysfunction and/or death of dopaminergic neurons. Restoration of dopaminergic neurons was shown in mice where β-catenin was activated in situ[52] and neural stem cells transplanted to the substantia nigra of medically PD-induced mice induced re-expression of Wnt1 and repair dopaminergic neurons[55]. DKK3 and PPP3CB were strongly downregulated in de novo PD, removing an important line of defence against the detrimental loss of dopaminergic neurons. The downregulation of the Wnt-signalling pathways was further correlated with higher motor scores (UDPRS and H&Y stages).

Wnt-signalling in PD is not only promising as a potential biomarker. In oncology, drugs can modify Wnt-pathways, which is of interest to the PD field[56]. Some substances show no BBB-permeability. As a disrupted BBB seems to be apparent in PD, these drugs may be effective. Furthermore, these substances are also relevant for PD treatment: research points towards a peripheral starting point of PD and future therapies should be administered as early as possible[57]. These promising substances include DKK- as well as GSK inhibitors, but to date, no drugs targeting the Wnt-signalling pathways have been effectively tested in clinical trials, including in those with neurodegenerative diseases. Progress and clinical trials are urgently needed here.

The transfer of multi-omics analysis to clinically meaningful results that directly impact future drug trial planning and biomarker validation, depends fundamentally on correlating these results and altered pathway regulations with established clinical scores. The markers we analysed in our targeted mass spectrometry panel did not only show different expression patterns between HC, PD, and in both of our independent iRBD sample sets, but most of the markers also robustly correlated with important clinical scores (UPDRS and MMSE, see Table 1). Cognitive decline correlated negatively with the SERPINs and complement factor C3. The burden of motor and non-motor symptoms and overall symptom severity rated by UPDRS and its subscores correlated positively with the SERPINs, Complement C3, and negatively with DKK3, GRN, and SELE. So, increased inflammatory activity and downregulation of Wnt-signalling seem to strongly affect the clinical picture of PD subjects.

The iRBD subjects showed decreased levels of BCHE over time compared to controls. BCHE has been reported as decreased in serum

samples of PD with cognitive impairment[58]. Validation of this easily assessable marker in serum is needed to evaluate its predictive potential.

While we did not find significant differences when we compared paired serum and plasma samples; the analysis of paired samples of plasma/serum and CSF only correlated weakly with the marker concentrations in these peripheral and central compartments. This discrepancy has been reported by several groups[20,21]. One reason is that mass spectrometry-based proteome analysis is always biased towards quantification and detection of the most abundant proteins in each sample matrix, and the total protein concentrations in human plasma/serum are more than two orders of magnitude higher than that in CSF. Further, the regulatory function of the blood–brain barrier seems to play a different role for different proteins, as some, like c-reactive protein, show a strong correlation between CSF and plasma, but most of the proteins do not. CSF and blood proteome show complex dynamics influenced by multiple and still mostly unknown factors. The protein shift in samples with a known BBB dysfunction (determined by the CSF/serum albumin index or the CSF/plasma ratio) can not be determined for individual proteins nor the dysfunction be localised by mass spectrometry[20].

Our model could not correctly predict phenoconversion in all cases. The reasons for this can be varied: The proteome pattern changes over time and the period between sampling and phenconversion may play a role. The three PD phenoconverters that were not predicted as PD neither differ clinically or demographically from the phenoconverters, nor from the non-phenoconverters. iRBD diagnosis in our study was confirmed by vPSG, supported by a high percentage of additional measurements including hyposmia and CSF SAA positivity. Therefore, even those iRBD cases that do not show the PD-proteome pattern still have a high-risk constellation of converting to PD/DLB on three different levels (PSG, olfaction, and SAA). Continuing further longitudinal follow-up of these subjects will elucidate our understanding of when and potentially why conversion occurs/does not occur. It is known that around 80% of iRBD subjects develop NSD, i.e. PD/DLB, with a rate of 6% per year, as shown in a multicenter cohort including ours[59]. To a lesser extent, iRBD subjects develop the intracytoplasmic glial α-synuclein aggregation disorder Multiple Systems Atrophy (MSA)[59,60]. Although RBD is common in MSA (summary prevalence of 73%[61]), none of our iRBD subjects have, as yet converted to MSA. Recruiting and following large longitudinal at-risk cohorts is, therefore, very important and future studies will not only identify biomarkers for phenoconversion from stage 1 or 2 to eventually stage 3 NSD or MSA, but also identify the many possible factors of resilience (including genetics, etc.) of NON-conversion which will be as, if not more important than identifying indicators for phenoconversion. Both direction progression biomarkers from stage 1 and 2 cohorts will have tremendous implications for future neuroprevention trials as phenoconversion itself is (due to the low annual rate) unlikely to be an outcome measure.

A significant strength of our biomarker discovery to translation pipeline is that it allows for the developed test to be easily validated and translated to any clinical laboratory equipped with a tandem LC-MS instrument. One advantage of using triple quadrupole platforms is that additional and better biomarkers can easily be augmented into the test described in this manuscript. Thus, any test could be refined and optimised over time with very little modification to the assay as additional biomarkers are discovered. Clinical testing for neurological disorders is limited to the use of a selected few well-characterised individual markers and translating biomarkers to eventual clinical application is notoriously challenging. The power of using multiplexed biomarker technologies with machine learning enables biomarkers to be evaluated in context with other markers of pathological events, thereby creating a 'disease profile' as opposed to individual markers. This approach opens the biomarker discovery field for many disorders

and increases the specificity and sensitivity of testing, as demonstrated in this study. The combination of multiplexed analysis of biomarker panels analysed on triple quadrupole platforms can advance biomarker translation to clinical application; this mass spectral technology is already embedded in many clinical diagnostics labs for routine small molecule analyses.

Our peripheral blood protein pattern for PD helps not only to classify but also to predict the earliest stage of the disease. We find differently expressed proteins in pre-motor iRBD and early motor stages of the disease compared to HC. Multiple markers also correlated with the progression of motor and non-motors symptoms. Thus, our blood panel can also identify subjects at risk (stage 2) to develop PD up to 7 years before advancing to motor stage 3. Next steps will be the independent validation in other (and even earlier) non-motor cohorts, e.g. in subjects with hyposmia also at-risk for PD [62] and in our population-based Healthy Brain Ageing cohort in Kassel[63]. It would further be interesting to evaluate the predictive potential of these identified markers with continuing clinical follow-up and together with other established PD progression markers like serum neurofilament light chain[5] and dopamine transporter imaging in a longitudinal analysis.

Our work was predominantly focused on the similarities between PD and iRBD. The authors are unaware of any study that has analysed longitudinally collected samples and prodromal cohorts, including iRBD and phenoconverters. Future work would include (i) validation of our findings in independent cohorts consisting of iRBD and other at-risk subjects for the synuclein aggregation disorders in neurons (PD, DLB) and oligodendrocytes (MSA), (ii) refinement of the panels of biomarkers developed in this study including sensitivity and technical performance, (iii) and using the pipeline described in this manuscript, the identification and validation of additional biomarkers that could distinguish between the different clinical syndromes with the ultimate goal of identifying progression biomarkers as outcome measures for prevention trials.

In summary, instead of single biomarkers, in a univariate approach, we have created a pipeline using a targeted proteomic test of a multiplexed panel of proteins, together with machine learning. This powerful combination of multiple well-selected biomarkers with state-of-the-art machine-learning bioinformatics, allowed us to use a panel of eight biomarkers that could distinguish early PD from HC. This biomarker panel provided a distinct signature of protective and detrimental mechanisms, finally triggering oxidative stress and neuroinflammation, leading to α-synuclein aggregation and LB formation. Moreover, this signature was already present in the prodromal non-motor (stage 2 NSD), up to 7 years before the development of motor/cognitive symptoms (stage 3), supporting the high specificity of iRBD and its high conversion rate to PD/DLB[18]. Most importantly, this blood panel can, in the future, upon further validation help identify subjects at risk of developing PD/DLB and stratify them for upcoming prevention trials.

## Methods

### Patient cohorts and sample collection and processing

Our research complies with all relevant ethical regulations. Institutional review board statements were obtained from the University Medical Centre in Goettingen, Germany, Approval No. 9/7/04 and 36/7/02. The study was conducted according to the Declaration of Helsinki, and all participants gave written informed consent. All plasma, serum and CSF samples from subjects were selected from known cohorts using identical sample processing protocols designed by the Movement Disorder Center Paracelsus-Elena-Clinic.

Patients with de novo PD were diagnosed according to the UK Brain Bank Criteria, without PD-specific medication. Diagnosis in all subjects was supported by (1) a positive (i.e. >30% improvement of UPDRS III after 250 mg of levodopa) acute levodopa challenge testing[64]

in all PD subjects, (2) hyposmia by smell identification test (Sniffin Sticks[65]) in all PD subjects and (3) 1.5-tesla Magnetic Resonance Imaging (MRI) without significant abnormalities or evidence for other diseases in all but three subjects who were excluded (due to significant vascular lesions or evidence for hydrocephalus) from the analysis. Participants not fulfilling the above criteria and meeting criteria for other neurological disorders were named as other neurological disorders (OND). OND consists of subjects with vascular parkinsonism ($n = 10$), essential tremor ($n = 7$), progressive supranuclear palsy; PSP ($n = 7$), multiple system atrophy; MSA ($n = 3$), corticobasal syndrome; CBS ($n = 2$), DLB ($n = 2$), drug-induced tremor ($n = 2$), dystonic tremor ($n = 2$), restless legs syndrome ($n = 1$), hemifacial spasm ($n = 1$), moto-neuron disease ($n = 1$), amyotrophic shoulder neuralgia ($n = 1$), and Alzheimer's disease ($n = 1$). The initial exploratory cohort consisted of ten PD subjects (8 men, mean age $67.1 \pm 10.6$) and ten healthy controls (5 men, mean age 65,7, SD ± 8,6.). For details, see Supplementary Table 3. The validation cohort included 99 PD subjects (49 men, mean age 66,1, SD ± 10,8), 36 healthy controls (20 men, mean age 63.7, SD ± 6,5.) and the described (see above) 41 OND subjects (29 men, mean age 70, SD ± 8.9. For details, see Supplementary Table 1. The prodromal validation cohort consisted of 54 patients with iRBD (27 men, mean age 67.5, SD ± 8.1, for details, see Supplementary Table 4). RBD was diagnosed with two nights of state-of-the-art vPSG. Samples from HC were selected from the DeNoPa cohort[10] and matched for age and sex with the PD patients, had to be between 40 and 85 years old, without any active known/treated CNS condition, and with a negative family history of idiopathic PD. Antipsychotic drugs were an exclusion criterion. The provided data for sex are based on self-report.

The paired sample analysis of CSF, plasma and serum was applied in samples from subjects with OND 7 men, mean age 74 years, SD ± 7; diagnosis: four Alzheimer's disease, three vascular Parkinsonism, one essential tremor, one multiple system atrophy one progressive supranuclear palsy).

Clinical assessments included the UPDRS subscores (parts I–III), the sum (UPDRS total score), and cognitive screening using the MMSE[10].

Plasma and serum samples for both cohorts were collected in the morning under fasting conditions using Monovette tubes (Sarstedt, Nümbrecht, Germany) for EDTA plasma and serum collection by venipuncture. Tubes were centrifuged at 2500×g at room temperature (20 °C) for 10 min and aliquoted and frozen within 30 min of collection at −80 °C until analysis[10,66]. Single-use aliquots were used for all analyses presented here. For further details, we refer to the following publication[67].

CSF was collected in polypropylene tubes (Sarstedt, Nümbrecht, Germany) directly after the plasma collection by lumbar puncture in the sitting position. Tubes were centrifuged at 2500×g at room temperature (20 °C) for 10 min and aliquoted and frozen within 30 min after collection at −80 °C until analysis. Before centrifugation, white and red blood cell counts in CSF were determined manually[10,66]. CSF β-amyloid 1–42, total tau protein (t-tau), phosphorylated tau protein (p-tau181) and neurofilament light chains (NFL) concentrations were measured by board-certified laboratory technicians, who were blinded to clinical data, using commercially available INNOTEST ELISA kits for the tau and Aβ markers (Fujirebio Europe, Ghent, Belgium) and the UmanDiagnostics NF-light® assay (UmanDiagnostics, Umeå, Sweden) for NFL. Total protein and albumin levels were measured by nephelometry (Dade Behring/Siemens Healthcare Diagnostics)[66].

For the α-synuclein seeding aggregation assay (αSyn-SAA) the CSF samples were blindly analyzed in triplicate (40 μL/well) in a reaction mixture (0.3 mg/mL recombinant α-Syn (Amprion [California, USA]; catalogue number S2020), 100 mM piperazine-N,N′-bis(2-ethane-sulfonic acid) (PIPES) pH 6.50, 500 mM sodium chloride, 10 μM thio-flavin T, and one bovine serum albumin (BSA)−blocked 2.4-mm silicon nitride G3 bead (Tsubaki-Nakashima [Georgia, USA]). Beads were

blocked in 1% BSA 100 mM PIPES pH 6.50 and washed with 100 mM PIPES pH 6.50. The assay was performed in 96-well plates (Costar [New York, USA], catalogue number 3916) using a FLUOstar Omega fluo-rometer (BMG [Ortenberg, Germany]). Plates were orbitally shaken (800 rpm for 1 min every 29 min at 37 °C). Results from the triplicates were considered input for a three-output probabilistic algorithm with sample labelling as "positive," "negative," or "inconclusive", based on the parameters: Maximum fluorescence (Fmax), time to reach 50% Fmax (T50), slope, and the coefficient of determination for the fitting were calculated for each replicate using a sigmoidal equation available in Mars data analysis software (BMG). The time to reach the 5000 relative fluorescence units (RFU) threshold (TTT) was calculated with a user-defined equation in Mars[19].

## Discovery plasma proteomics (phase 0)

In the mass spectrometry-based proteomic discovery analysis of plasma, we depleted the control and de novo PD samples from the twelve most abundant plasma proteins using Pierce Top12 columns (Thermo Fisher Scientific, Waltham, MA, USA) according to the manufacturer's instructions. The depleted samples were freeze-dried before the addition of 20 μL of lysis buffer (100 mM Tris pH 7.8, 6 M urea, 2 M thiourea, and 2% ASB-14). The samples were shaken on an orbital shaker for 60 min at 1500 rpm. To break disulphide bonds, 45 μg DTE was added, and the samples were incubated for 60 min. To prevent disulphide bonds from reforming, 108 μg IAA was added, and the samples were incubated for 45 min covered in light. About 165 μL MilliQ water was added to dilute the concentration of urea and 1 μg trypsin gold (Promega, Mannheim, Germany) was added before 16 h of incubation at +37 °C to digest the proteins into peptides. To purify the peptides, solid phase extraction was performed using 100 mg C18 cartridges (Biotage, Uppsala, Sweden). The cartridges were washed with two 1 mL aliquots of 60% ACN, and 0.1% TFA before equilibration by two 1 mL aliquots of 0.1% TFA. The concentration of TFA in the samples was adjusted to 0.1%. The samples were loaded, and the flow-through was captured and re-applied. Salts were washed away from the bound peptides by two 1 mL aliquots of 0.1% TFA. The peptides were eluted by two 250 μL aliquots of 60% ACN, and 0.1% TFA. Solvents were evaporated using a vacuum concentrator. The samples were re-suspended in 50 μL 3% ACN, 0.1% FA prior to analysis. About 4 μL was injected into a 2D-NanoAquity liquid chromatography system (Waters, Manchester, UK). All samples were fractionated online into ten fractions over 12 h. The mobile phase in the first chromatographic system consisted of A1: 10 mM ammonium hydroxide titrated to pH 9 and B1: acetonitrile. The second chromatographic system's mobile phase was A2: 5% dimethylsulfoxide (DMSO) + 0.1% formic acid, B2: acetonitrile with 5% DMSO + 0.1% formic acid. 2D-liquid chromatography fractio-nation was performed by loading the sample onto a 300 μm × 50 mm, 5 μm Peptide BEH C18 column (Waters). The peptides were eluted from the first column at a flow rate of 2 μL/min. The initial condition of the gradient elution was 3% B, held over 0.5 minutes before linearly increasing the proportion of organic solvent B, fraction per fraction over 0.5 min. The conditions thereafter remained static for 4 min before returning to the initial conditions over 0.5 min and equilibra-tion prior to the next elution for 10 min. The eluted peptides from the first-dimensional column were loaded into a 180 μm × 20 mm, 5 μm Symmetry C18 trap column (Waters) before entering the analytical column, a 75 μm × 150 mm, 1.7 μm Peptide BEH C18 (Waters). The col-umn temperature was +45 °C. The gradient elution applied to the analytical column started at 3% B and was linearly increased to 40% B over 40 min after which it was increased to 85% B over 2 min and washed for 2 min before returning to initial conditions over 2 min followed by equilibration for 15 min before the subsequent injection. The eluted peptides were detected using a Synapt-G2-Si (Waters) equipped with a nano-electrospray ion source. Data were acquired in positive MS$^E$ mode from 0 to 60 min within the m/z range 50–2000.

The capillary voltage was set to 3 kV and the source temperature to +100 °C. The desolvation gas consisted of nitrogen with a flow rate of 50 L/h, and the desolvation temperature was set to +200 °C. The purge and desolvation gas consisted of nitrogen, operated at a flow rate of 600 mL/h and 600 L/h, respectively. The gas in the IMS cell was helium, with a flow rate of 90 mL/h. The low energy acquisition was performed by applying a constant collision energy of 4 V with a 1-s scan time. High energy acquisition was performed by applying a collision energy ramp, from 15 to 40 V, and the scan time was 1 s. The lock mass consisted of 500 fmol/μL [glu1]-fibrinopeptide B, continuously infused at a flow rate of 0.3 μL/min and acquired every 30 s. The doubly charged precursor ion, m/z 785.8426, was utilised for mass correction. After acquisition, data were imported to Progenesis QI for proteomics (Waters), and the individual fractions were processed before all results were merged into one experiment. The Ion Accounting workflow was utilised, with UniProt Canonical Human Proteome as a database (build 2016). The digestion enzyme was set as trypsin. Carbamidomethyl on cysteines was set as a fixed modification; deamidation of glutamine and asparagine, and oxidation of tryptophan and pyrrolidone carboxylic acid on the N-terminus were set as variable modifications. The identification tolerance was restricted to at least two fragments per peptide, three fragments per protein, and one peptide per protein. A FDR of 4% or less was accepted. The resulting identifications and intensities were exported and variables with a confidence score less than 15 and only one unique peptide were filtered out.

### Targeted plasma proteomics (phase I)

The peptides included in the targeted assay were selected from several proteomic screening studies in which we analysed plasma, serum, urine, and CSF in ageing, PD and AD. The analytical method is described by ref. 17. Furthermore, due to the suggested involvement of inflammation in neurodegenerative diseases, several known pro- and anti-inflammatory proteins identified from the literature were included in the multiplexed assay. The final panel consisted of 121 proteins (Supplementary Table 2), out of which a number were measured with two peptides, leading to a total of 167 unique peptides. When possible, the peptides were chosen to have an amino acid sequence length between 7 and 20. The amino acid sequences were confirmed to be unique to the proteins by using the Basic Local Alignment Search Tool (BLAST) provided by UniProt[68]. Synthetic peptide standards were purchased from GenScript (Amsterdam, Netherlands). To establish the most optimal transitions, repeated injections of 1 pmol peptide standard onto a Waters Acquity ultra-performance liquid chromatography (UPLC) system coupled to a Waters Xevo-TQ-S triple quadrupole MS were performed. The most high-abundant precursor-to-product ion transitions and their optimal collision energies were determined manually or using Skyline[69]. Detection was performed in positive ESI mode. The capillary voltage was set to 2.8 kV, the source temperature to 150 °C, the desolvation temperature to 600 °C, and the cone gas and desolvation gas flows to 150 and 1000 L/h, respectively. The collision gas consisted of nitrogen and was set to 0.15 mL/min. The nebuliser operated at 7 bar. Two transitions were chosen, one quantifier for relative concentration determination and one qualifier for identification, totally rendering 334 analyte transitions. Cone and collision energies varied depending on the optimal settings for each peptide. Each peptide was measured with a minimum of 12 points per peak and a dwell time of 10 ms or more to ensure adequate data acquisition. The optimised transitions were distributed over two multiple reaction monitoring (MRM) methods, always keeping the quantifier and qualifier for each peptide in the same MRM segment. Plasma, serum, and CSF samples were depleted from albumin and IgG using Pierce Top2 cartridges (Thermo Fisher Scientific, Waltham, MA, USA) following the manufacturer's instructions. About 150 μg whole protein yeast enolase (ENO1) was added to the cartridges as an internal standard to account for digestion efficiency. Digestion was performed as described above.

Solid phase extraction was carried out on BondElute 100 mg C18 96-well plates (Agilent, Santa Clara, USA) using the same methodology as in the preparation of untargeted proteomic analyses. Quality control samples were prepared from acetone-precipitated plasma, digested and solid phase extracted. Calibration curves ranging from 0 to 1 pmol/μL were constructed in blank and matrix by spiking increasing amounts of peptides into blank and QC samples. Before analysis, the samples were reconstituted in 30 μL 3% ACN, 0.1% FA containing 0.1 μM heavy isotope labelled peptides from the following proteins (annotated by gene name): ALDOA, C3, GSTO1, RSU1 and TSP1. About 5 μL were injected. The peptides were separated and detected on an Acquity UPLC system coupled to a Xevo-TQ-S triple quadrupole mass spectrometer (Waters, Manchester, UK). Chromatographic separation of the peptides was performed using a 1 × 100 mm, 1.7 μm ACQUITY UPLC Peptide CSH C18 column (Waters).

The mobile phase consisted of A: 0.1% formic acid and B: 0.1% formic acid in acetonitrile pumped at a flow rate of 0.2 mL/min. The column temperature was set to +55 °C. The initial mobile phase composition was 3% B, which was kept static for 0.8 min before initialising the linear gradient, running for 7.6 min to 25% B, eluting most of the peptides. B was thereafter linearly increased to 80% over 0.5 min and held for 1.9 min, eluting the most apolar peptides and washing the column before returning to the initial conditions over 0.1 minutes followed by equilibration for 6 min prior to the subsequent injection. Two subsequent injections of each sample were performed, each paired with one of the two MRM acquisition methods.

After acquisition, peak-picking and integration were performed using TargetLynx (version 4.1, Waters) or an in-house application ('mrmIntegrate') written in Python (version 3.8). mrmIntegrate is publicly available to download via the GitHub repository https://github.com/jchallqvist/mrmIntegrate. The application takes text files as input (.raw files are transformed into text files through the application 'MSConvert' from ProteoWizard[70] and applies a LOWESS filter over five points of the chromatogram. The integration method to produce areas under the curve is trapezoidal integration. The application enables retention time alignment and simultaneous integration of the same transition for all samples. Peptide peaks were identified by the blank and matrix calibration curves. The integrated peak areas were exported to Microsoft Excel, where first, the ratio between quantifier and qualifier peak areas were evaluated to ensure that the correct peaks had been integrated. The digestion efficiency was evaluated by monitoring the presence of baker's yeast ENO1 in the samples, all samples without a signal were excluded from further analysis. After the initial quality assessment, the quantifier area was divided by the area of one of the internal standards, ALDOA or GSTO1 to yield a ratio used for the determination of relative concentrations. Any compound that also showed an intensity signal in the blank samples had the blank signal subtracted from the analyte peak intensity. Pooled plasma quality control samples were additionally evaluated to assess the robustness of the run.

### Refined LC-MS/MS method (phase II)

The rapid and refined targeted proteomics LC-MS/MS method contained only peptides from the 31 proteins observed in the original targeted proteomics method (121 proteins). We utilised a Waters Acquity (UPLC) system coupled to a Waters Xevo-TQ-XS triple quadrupole operating in positive ESI mode. The column was an ACQUITY Premier Peptide BEH C18, 300 Å, 1.7 μm, maintained at 40 °C. The mobile phase was A: 0.1% formic acid in water, and B: 0.1% formic acid in acetonitrile. The gradient elution profile was initiated with 5% B and held for 0.25 min before linearly increasing to 40% B over 9.75 min to elute and separate the peptides. The column was washed for 1.6 min with 85% B before returning to the initial conditions and equilibrating for 0.4 min. The flow rate was 0.6 mL/min. The settings of the mass spectrometer and the peak-picking method were the same as

described in the prior section. Baker's yeast ENO1 was utilised to monitor digestion efficiency and as an internal standard.

## Statistical methods

Most of the statistical analyses were performed in Python (version 3.8.5). The untargeted and targeted datasets were inspected for outliers and instrumental drift using principal component analysis (PCA) and orthogonal projection to latent variables (OPLS) in SIMCA, version 17 (Umetrics Sartorius Stedim, Umeå, Sweden). Outliers exceeding ten median deviations from each variable's median were excluded. Instrumental drift was corrected by applying a non-parametric LOWESS filter from statsmodels (version 0.14.0) using 0.5 fractions of the data to estimate the LOWESS curve[71]. The data were evaluated for normal distribution using D'Agostino and Pearson's method from SciPy (version 1.9.3)[72]. The non-normally distributed variables in the untargeted data were transformed to normality by the Box-Cox procedure using the SciPy function 'boxcox'. Significance testing between the independent groups of HC and PD/OND/iRBD individuals was performed by Student's two-tailed $t$-test for the untargeted proteomic data and by Mann–Whitney's non-parametric $U$-test (SciPy) for the targeted data. Due to the limited sample numbers, no multiple testing correction was performed in the untargeted data. In the targeted data, the Benjamini–Hochberg multiple testing correction procedure (statsmodels) was applied with an accepted false discovery rate of 5%. Fold-changes were calculated by dividing the means of the affected groups by the control group. Correlation analyses in the targeted data were performed by Spearman's correlation (SciPy) and the correlation $p$ values were adjusted variable-wise by the Benjamini–Hochberg procedure (FDR = 5%).

We implemented a support vector classifier model to discriminate between PD and HC and to predict new samples. The data were first z-scored protein-wise and any 'not a number'-values were replaced by the median. We used the 'LinearSVC' method from SciKit Learn and applied cross-validated recursive feature elimination to determine the number of variables to use in the model. The most discriminating variables for distinguishing between controls and PD were thereafter chosen by recursive feature elimination[73]. Feature selection and model training were performed on 70% of the data, partitioned using the SciKit Learn function "train_test_split", and cross-validation was performed using a stratified k-fold with five splits. The remaining 30% of the data were predicted in the model. PR and ROC curves were constructed from the test data and consisted of each predictor and from the combined predictors, the packages precision_recall_curve and roc_curve from SciKit Learn were implemented. Linear mixed models were performed using the R-to-Python bridge software pymer4 (version 0.8.0), where individual was set as a random effect and the correlations between the MS measured proteins and clinical variables were evaluated for significance post Benjamini–Hochberg's procedure for multiple testing correction. Plots of the data were constructed using the Seaborn and Matplotlib packages (versions 0.12.2 and 3.6.0, respectively)[74].

All multivariate analyses were performed in SIMCA, version 17. OPLS and OPLS-discriminant analysis (OPLS-DA) models were evaluated for significance by ANOVA $p$ values and by permutation tests applying 1000 permutations, where $p < 0.05$ and $p < 0.001$ were deemed significant, respectively.

Data were analysed for pathway enrichment using IPA (QIAGEN Inc. Data were analysed for pathway enrichment using IPA (QIAGEN Inc., https://digitalinsights.qiagen.com/products-overview/discovery-insights-portfolio/analysis-and-visualization/qiagen-ipa/.). Input variables were set to proteins demonstrating a significant difference between PD individuals and HC, with fold-change as expression observation. The accepted output pathways were restricted to $p < 0.05$ and at least two proteins were included in the pathways. Gene Ontology (GO) annotations were extracted using DAVID Bioinformatics

Resources (2021 build)[75,76]. Networks were built in Cytoscape[77] (version 3.8.0) by applying the "Organic layout" from yFiles[77].

## Obtaining biological materials

Patient samples can be provided to other researchers for certain projects after contact with the corresponding authors and upon availability approval of the team in Kassel, Germany.

## Reporting summary

Further information on research design is available in the Nature Portfolio Reporting Summary linked to this article.

## Data availability

The chromatograms from the targeted mass spectrometric data generated in this study have been deposited in the ProteomeXchange database under accession code PXD041419 and in the Panorama repository (https://panoramaweb.org/DNP_Pub.url, https://doi.org/10.6069/p9cy-h335). The integrated targeted mass spectrometric data generated in this study are provided in the Supplementary Information. Source data for all data presented in graphs within the figures are provided in a source data file. Source data are provided with this paper.

## Code availability

Peak-picking and integrations were performed in TargetLynx (part of the MassLynx suite, version 4.1), or using an in-house application written in Python which can be found on GitHub (https://github.com/jchallqvist/mrmIntegrate). The data visualisation and statistical analyses were performed in Python (version 3.8.5) using the packages SciPy (version 1.9.3), statsmodels (version 0.14.0), SciKit Learn (version 1.1.2), Seaborn (version 13.0) and Matplotlib (version 3.6.0). The code used can be found on GitHub (https://github.com/jchallqvist/DNP_Pub/blob/main/DNP_Code, https://doi.org/10.5281/zenodo.11130369).

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

## Acknowledgements
This work was supported by the Michael J Fox Foundation, PDUK, The Peto Foundation, The TMSRG (UCL), The BRC at Great Ormond Street Hospital, and the Horizon 2020 Framework Programme (Grant number 634821, PROPAG-AGING). We thank the PROPAG-AGING consortium, a full list of the members can be found in the supplementary material.

## Author contributions
J.H., M.B., K.M., and B.M. conceptualised, planned and oversaw all aspects of the study. J.H., K.M., J.S., H.V., M.B. and S. Schreglmann performed and analyzed most of the experiments. S. Schade, S.W. and M.B. consented to the subjects and collected the samples. M.-L.M., F.S.-D. and S. Schade analyzed the sleep lab data and diagnosed the iRBD subjects. J.H. and M.D. performed the statistical data analysis. J.H. applied the machine learning methods and designed the figures. W.H., I.D., C.F., M.-G.B., P.G., C.P., K.B. and M.X. provided substantial contributions to the conception of the work, acquisition and interpretation of the data, particularly for the mass spectrometry setup and the refinement of the targeted panel. S. Schade, S.W., C.T., M.B., B.M., M.-L.M. and F.S.D. conceptualised the clinical study, analyzed the clinical data and reevaluated the diagnosis. M.E. provided substantial contributions to the clinical data analyzes, particularly the imaging patient data in regard to differential diagnosis. J.H., M.B., K.M. and B.M. wrote the manuscript with input and substantial revisions from all authors.

## Funding

## Competing interests
JH, MD, MX, SW, KB, ME, PG, MGB, CP, KM, ID, WH, JS, HV and CF and have no competing interests to report. MB has received funding from the Deutsche Forschungsgemeinschaft (DFG, German Research Foundation) – 413,501,650. CT has received honoraria for consultancy from Roche, and honoraria for educational lectures from UCB, and has received research funding for the PPMI study from the Michael J. Fox Foundation and funding from the EU (Horizon 2020) and stipends from the (International Parkinson's and Movement Disorder Society) IPMDS. BM has received honoraria for consultancy from Roche, Biogen, AbbVie, UCB, and Sun Pharma Advanced Research Company. BM is a member of the executive steering committee of the Parkinson Progression Marker Initiative and PI of the Systemic Synuclein Sampling Study of the Michael J. Fox Foundation for Parkinson's Research and has received research funding from the Deutsche Forschungsgemeinschaft (DFG), EU (Horizon 2020), Parkinson Fonds Deutschland, Deutsche Parkinson Vereinigung, Parkinson's Foundation and the Michael J. Fox Foundation for Parkinson's Research. MLM has received honoraria for speaking engagements from Deutsche Parkinson Gesellschaft e.V., and royalties from Gesellschaft fur Medien + Kommunikation mbH + Co. FSD has received honoraria for speaking engagements from AbbVie, Bial, Ever Pharma, Medtronic and royalties from Elsevier and Springer. She served on an advisory board for Zambon and Stada Pharma. FSD participated in Ad Boards for consultation: Abbvie, UCB, Bial, Ono, Roche and got honorary for lecturing: Stada Pharm, AbbVie, Alexion, Bial. S. Schade received institutional salaries supported by the EU Horizon 2020 research and innovation programme under grant agreement No. 863664 and by the Michael J. Fox Foundation for Parkinson's Research under grant agreement No. MJFF-021923. He is supported by a PPMI Early Stage Investigators Funding Programme fellowship of the Michael J. Fox Foundation

for Parkinson's Research under grant agreement No. MJFF-022656. S. Schreglmann received institutional salaries supported by the EU Horizon 2020 research and innovation programme under grant agreement No. 863664, support from the Advanced Clinician Scientist programme by the Interdisciplinary Centre for Clinical Research, Wuerzburg, Germany, and from the Deutsche Forschungsgemeinschaft (DFG, German Research Foundation) Project-ID 424778381-TRR 295. He is a fellow of the Thiemann Foundation. He serves as a scientific adviser to Ele-mind Inc.

## Additional information

[1]UCL Institute of Child Health and Great Ormond Street Hospital, London, UK. [2]UCL Queen Square Institute of Neurology, Clinical and Movement Neurosciences, London, UK. [3]Department of Neurology, University Medical Center Goettingen, Goettingen, Germany. [4]Institute for Neuroimmunology and Multiple Sclerosis Research, University Medical Center Goettingen, Goettingen, Germany. [5]Paracelsus-Elena-Klinik, Kassel, Germany. [6]Department of Experimental, Diagnostic, and Specialty Medicine (DIMES), University of Bologna, Bologna, Italy. [7]IRCCS Istituto delle Scienze Neurologiche di Bologna, Bologna, Italy. [8]National Hospital for Neurology & Neurosurgery, Queen Square, WC1N3BG London, UK. [9]Institute of Diagnostic and Interventional Neuroradiology, University Medical Center Goettingen, Goettingen, Germany. [10]Department of Neurology, Philipps-University, Marburg, Germany. [11]UCL: Food, Microbiomes and Health Institute Strategic Programme, Quadram Institute Bioscience, Norwich Research Park, Norwich, UK. [12]Department of Neurosurgery, University Medical Center Goettingen, Goettingen, Germany. [13]These authors contributed equally: Jenny Hällqvist, Michael Bartl. [14]These authors jointly supervised this work: Kevin Mills, Brit Mollenhauer. ✉e-mail: j.hallqvist@ucl.ac.uk; michael.bartl@med.uni-goettingen.de

