## [Peer Review File · Nature Communications]

Plasma proteomics identify biomarkers predicting Parkinson's disease up to 7 years before symptom onsetREVIEWER COMMENTS

Reviewer #1 (Remarks to the Author):

Using LC/MS-based proteomics approach, authors developed the plasma biomarker panel to identify Parkinson's disease (PD) and the prodromal symptom of isolated REM sleep behavior disorder (iRBD). Authors eventually evaluated 18 prodromal subjects with iRBD and predicted 72 - 94% of the iRBD samples as PD, which matches the clinical conversion rate observed in PD, identifying a pattern already evident in iRBD and indicating pre-symptomatic molecular events. Here, I saw several critical points described below are missing in the current manuscript to conclude author's statement that the panel can be employed to detect prodromal and early Parkinson's disease (prognosis, diagnosis, and monitoring).

1. First, since the pixel quality of figures is too low in this manuscript, it was so difficult to see all the results in figures. Need to improve the visibility of the results in all figures. In the current manuscript, I could not see the details at all.
2. Authors analyzed 18 plasma samples from prodromal subjects with iRBD and predicted 72 - 94% of the iRBD samples as PD. Authors mentioned this matches the clinical conversion rate observed in PD generally, identifying a pattern already evident in iRBD and indicating pre-symptomatic molecular events. To make this impactful statement, it is suggested that the longitudinal analysis should be conducted to see the true conversion of the iRBD patients with the "developed biomarker panel – positive" to the PD onset. It is necessary to confirm e.g., 72-94% of the individual iRBD patients predicted as prodromal PD by the developed biomarker panel are certainly true. Unless this step is conducted, it is difficult to assess the biomarker performance for the prognosis viewpoint.
3. In the current manuscript, all discussions to identify PD patients were conducted by "plasma biomarkers". I agree with that the establishing plasma biomarker panel is the ultimate goal with considering the clinical accessibility, however the investigations of the "CSF and brain tissues" (hopefully, paired with the plasma samples used for the biomarker panel construction) are also quite important to understand the mechanism of action for the plasma biomarker panel and increasing the reliability. Parkinson's disease is the CNS-disease, and the alpha-synuclein pathology as well as the damage on dopaminergic neurons are observed in brain parenchyma, not peripheral blood. Here, authors need to add the data and major discussion about the mechanism link between the peripheral biomarker changes and CNS-PD pathology and neurodegeneration. To do that, it is strongly recommended to add the data on CSF and brain tissues (preferably, both the positive and negative control brain regions) proteomics data using the same discovery non-biased method approach.
4. Authors stated the early detection of PD at even prodromal stage - iRBD can be achieved by the established plasma biomarker panel consisted of the proteins relevant to the inflammation mainly. Indeed, some reports suggest that the inflammation may precede the formation of alpha-synuclein pathology, which implies that the pathogenesis of alpha-synuclein aggregates may be driven by inflammatory pathway. However, this hypothesis is still under discussion, and the authors need to clarify if the developed plasma biomarker panel is proximally relevant to the severity of alpha-synuclein pathology, or just relevant to the PD clinical symptom as clinical phenotype. To clarify this point, one

suggested option is that the analysis of pathologically-confirmed (alpha-synucleinopathy with different Braak/Spreading stages) plasma samples analysis, not just the clinically-diagnosed plasma samples. As another option, it is recommended to analyze the PD-model mice - e.g., Line61 etc. with progressing alpha-synuclein pathology in time-course. Even non-clinical studies are also helpful to characterize/understand the mode-of action for the established clinical biomarkers – as a traditional approach.

5. In Line 195-, authors mentioned, “We additionally predicted the OND samples, out of which nine were predicted as HC, 12 as PD, and 19 were not predicted as belonging to either group. The 12 samples predicted as PD did not demonstrate enrichment in any of the clinical OND groups. The random distribution of the OND samples between PD and HC indicates that the heterogeneous group of OND individuals does not share a distinct protein expression with either the HC or PD groups.” Here, it is necessary to further discuss about the specificity of the developed plasma assay panel for PD (or alpha-synucleinopathy?). For instance, 12 OND participants were classified as PD by the developed plasma biomarker panel, then if so, are there some specific characteristics in these 12 OND participants – e.g., the DLB participants with alpha-synuclein aggregates are enriched in this specific group identified as PD by the developed biomarker panel?

6. In the current manuscript, the bidirectional performance to diagnose e.g., HC vs PD by the developed biomarker panel is mainly discussed. Here, authors should investigate if the developed panel can be applied to assess the disease severity in terms of the clinical scales and also the amount of pathology – i.e., alpha-synuclein aggregates in brain. In Supplementary Table 2, some data for Spearman correlations between each protein biomarker and clinical scales are described, however it seems that all individual rho values are very low and difficult to be used to predict the severity in a quantitative manner. Even though the individual proteins do not show the convincing results on predicting the disease severity, some regression models with composite score can be constructed for the quantitative assessment, not the bidirectional diagnosis purpose only?

Reviewer #2 (Remarks to the Author):

The authors sought to investigate whether a panel of plasma proteomic biomarkers could discriminate PD from healthy controls, and also be applicable to iRBD.

There are many positive aspects of this analysis and manuscript, which the authors describe very well. The manuscript is also well-written, and the figures and tables are informative.

Please address the following points:

The authors state that “Parkinson’s disease (PD) is an increasingly prevalent neurodegenerative disease” – please provide references that PD is increasing in prevalence.

Note the error in lines 142-143 – “Further details can be found in Supplementary Error! Reference source

not found”

In the discussion, considering the recent associations of GRN mutations, lysosomal function, Lewy body dementia, etc (for example, see Reho et al, *Mov Disord* 2022), it would be worth adding comments and associated references on this topic.

This analysis obviously focused on proteomic plasma biomarkers in PD. The authors should comment on the findings as they related to dementia with Lewy bodies (DLB) and MSA. Comments would be particularly needed to explain the findings in the small iRBD cohort. The panel predicted 72-94% of the iRBD samples as PD, which could be due to several factors/issues. One could be the possible change in the natural evolution of proteins from the prodromal (eg, iRBD) to full PD clinical syndrome phase, which the authors allude to. Another explanation is that some of the iRBD patients will evolve into a full DLB or MSA clinical syndromes. The proteomic panel may or may not be different between PD and DLB, and the authors should compare and contrast their findings with those of O’Bryant et al, *Alz and Dem* 2019. While one might predict that the panel would be different between PD and MSA, that is also an open question, which should also be discussed.

Reviewer #3 (Remarks to the Author):

See attached file.

[Editorial Note: The PDF is displayed over the next three pages]

Review “Proteomics and machine learning identify a distinct biomarker panel to detect prodromal and early Parkinson’s disease”

Summary

In their study, Hällqvist et. al. present a panel of proteomic plasma biomarkers that are capable of discriminating PD and healthy controls with perfect accuracy in a machine learning model. They apply their model on prodromal subjects with iRBD and predict samples with a similar rate as expected from clinical conversion rates.

Overall, the study is well-written and concise. The figures are informative. Mass spectrometry-based plasma proteomics to uncover biomarker panels has been studied for several years now. Although cerebrospinal fluid (CSF) is often the preferred body fluid for Parkinson’s research, plasma is preferred due to its potential for clinical applications, owing to its ease of access.

In current studies, a machine learning (ML) layer is often added following the identification of proteins of interest. This layer serves to build predictive models, a process that is becoming increasingly standardized. Sometimes, these studies correlate clinical parameters with protein values to provide deeper insights. Novel studies in this field usually leverage recent technological advancements to achieve greater protein depth, involving larger cohorts for increased reliability, or utilizing innovative algorithms for data interpretation. An early example reference would be Pan et al. (Journal of Proteome Research 2014, <https://doi.org/10.1021/pr500421v>), which used targeted plasma proteomics to study biomarkers generated from CSF measurements on a cohort of 282 patients, achieving an AUC of 0.753 with four peptides with a linear model.

In evaluating the current study, however, there seems to be a lack of distinctive technological advancement, an unusually large cohort, or an innovative algorithm. There doesn’t appear to be any significant leap from the existing state-of-the-art, based on the manuscript text. Yet, this should not overshadow the development of a protein panel that exhibits high predictive power and can achieve impeccable accuracy. Such an achievement could indeed be regarded as a form of novelty that is of great interest for the community.

Major: Study Design

For the above reasons, I find it extremely important how this predictive panel was generated. The authors conduct a proteomics discovery study and identify 47 proteins as differentially expressed. However, their final multiplexed assay consists of 121 proteins, stemming from “unpublished discovery studies”, AD, and proteins in the literature. The referenced literature includes an Alzheimer’s mouse model from 2005 and a study of Neuroinflammation in Schizophrenia. Essentially, there is a well-defined statistical framework for 47 proteins, and then 74 more are added in an opaque way. This, however, invalidates the whole discovery approach and severely limits the biological significance of the subsequent targeted analysis.

To address this, I would recommend the following:

- State how the final list of proteins was generated and included a list of them. In line 502, it is referenced to be Supplementary Table 2, but this is the correlation to clinical parameters. It is evident that the community will derive little value if the highly predictive panel is not disclosed.

- Conduct a comparison between the significant proteins from the discovery phase and the proteins featured in the targeted assay. There seems to be a discrepancy, as some of the highly regulated proteins from Supplementary Figure S1 are not present in Figure 1. What is the authors' explanation for this? Maintaining consistency in the representation of data, such as using either lollipop charts or volcano plots (or ideally both), would facilitate interpretation.
- Enhance the characterization of the identified proteins. While a machine learning model provides a quantitative metric of model performance, it would be beneficial to see additional technical controls. These could include correlation plots with hierarchical clustering and box/swarm plots for relevant proteins.

Major: ML model

The characterization of the machine learning (ML) model in this study could benefit from more thorough detailing. A receiver operating characteristic (ROC) curve based solely on the training set doesn't provide a comprehensive understanding of the model's generalizability. This analysis should be conducted on the test set. Additionally, the report lacks standard classification metrics such as precision, recall, and F1 score. Given that the dataset is imbalanced, with 99 Parkinson's Disease (PD) cases versus 36 healthy controls, a model that categorically predicts PD would already achieve 73% accuracy. Therefore, balanced accuracy or the Matthews correlation coefficient (MCC) should be considered for a more accurate evaluation. Alongside the ROC curve, a precision-recall (PR) curve would be a valuable additional metric.

The section spanning lines 581-585 is somewhat ambiguous. Was the 5-fold split applied to the 70% of the data? Without access to the ML code, it's difficult to fully understand the processes involved. While it's commendable that the feature extraction step was cross-validated, given the limited sample size, it would be very beneficial to also cross-validate the entire pipeline, rather than relying on a single split—ideally repeatedly. Scikit-learn's RepeatedKFold could be readily implemented for this purpose.

I found the correlation to clinical scores interesting, but, also confusing: On one hand, the aim is to develop a predictive biomarker panel, while on the other, it's desirable to identify biomarkers that correlate with the scale. Wouldn't it make more sense to directly construct an ML model capable of predicting, for example, the clinical rating scale?

Protein panel validation

The authors could certainly benefit from providing a more thorough analysis of the significant proteins identified during both the discovery and targeted phases. Presently, there appears to be a degree of inconsistency in terms of the significant proteins identified, and this is somewhat perplexing. A comparison with existing literature could provide valuable insights and potentially explain these discrepancies.

The authors do this partly by comparing pathways, and, e.g., comparing to their previous studies, but when comparing to the volcano plot of their OLINK study there seem to be entirely different proteins. This could be done by providing a table with proteins and references where they were identified previously.

Misc:

- The auto-generated PDF appears to have formatting issues, e.g. Reference source not found, figures are not readable
- Figure 2: The figure contains long descriptions for the Proteins, whereas the text has short names. It would increase readability to stick to one.
- L398: Clinical testing for neurological disorders is limited to use of a select few well characterised individual markers and translating new biomarkers to eventual clinical application is notoriously challenging. “to use of a”

In conclusion, I cannot endorse this study for publication in its current form in Nature Communications. However, I believe that a highly predictive biomarker panel, as presented in this study, could be of considerable interest to the scientific community. Therefore, I would support the publication of this study, provided the necessary revisions are made.

Rebuttal letter for manuscript NCOMMS-23-14866

titled

Proteomics and machine learning identify a distinct blood biomarker panel to detect Parkinson's disease up to 7 years before motor disease

We thank the editors for their patience and reviewers for their valuable comments in regard of our manuscript. We have taken on-board all the points, suggestions and completed all the requests for more experiments. Some of the experimental analyses required the acquisition of unique sample sets which took time to collect and analyze. Subsequently, the manuscript has been revised accordingly and the addition of the extensive new data to the study has made the work significantly better. We would like to take this opportunity to thank the reviewers for their comments and suggestions.

In the following, we are answering the comments point-by-point.

Reviewer comments are displayed in black, the author response in blue, quotes from the manuscript in italics.

REVIEWER COMMENTS

Reviewer #1 (Remarks to the Author):

Using LC/MS-based proteomics approach, authors developed the plasma biomarker panel to identify Parkinson's disease (PD) and the prodromal symptom of isolated REM sleep behavior disorder (iRBD). Authors eventually evaluated 18 prodromal subjects with iRBD and predicted 72 - 94% of the iRBD samples as PD, which matches the clinical conversion rate observed in PD, identifying a pattern already evident in iRBD and indicating pre-symptomatic molecular events. Here, I saw several critical points described below are missing in the current manuscript to conclude author's statement that the panel can be employed to detect prodromal and early Parkinson's disease (prognosis, diagnosis, and monitoring).

Reviewer comment: 1. First, since the pixel quality of figures is too low in this manuscript, it was so difficult to see all the results in figures. Need to improve the visibility of the results in all figures. In the current manuscript, I could not see the details at all.

Our answer: We apologize for any inconvenience caused and submitted all images in high resolution in the revised version.

Reviewer comment: 2. Authors analyzed 18 plasma samples from prodromal subjects with iRBD and predicted 72 - 94% of the iRBD samples as PD. Authors mentioned this matches the

clinical conversion rate observed in PD generally, identifying a pattern already evident in iRBD and indicating pre-symptomatic molecular events. To make this impactful statement, it is suggested that the longitudinal analysis should be conducted to see the true conversion of the iRBD patients with the “developed biomarker panel – positive” to the PD onset. It is necessary to confirm e.g., 72-94% of the individual iRBD patients predicted as prodromal PD by the developed biomarker panel are certainly true. Unless this step is conducted, it is difficult to assess the biomarker performance for the prognosis viewpoint.

Our answer: We thank the reviewer for this important point and fully agree. Therefore, we acquired and analyzed with 146 samples significantly more samples and we can confirm, that 16 out of these longitudinal 54 subjects iRBD converted to a synucleinopathy/neuronal α -synuclein disease (phenoconverters), including clinical Parkinson's disease (11 cases) and Dementia with Lewy bodies (5 cases). Further, with our refined and simplified discriminant support vector machine model, we found that 79% of these samples were predicted as PD in the SVM models. In total, 27 of the 40 individuals with follow up samples had their baseline sample and their longitudinal follow-up samples predicted as PD. 10 out of 16 phenoconverters were classified as PD at all timepoints. We are pleased to report that this longitudinal dataset validates and supports our initial findings and prediction rates for phenoconversion.

We added the following parts to the manuscript:

Page 6/7, line 258-283, main manuscript: Development of a rapid and refined LC-MS/MS method and evaluation of a larger iRBD cohort, with longitudinal samples (Phase II)

*“To evaluate the results from the initial prediction models focusing on at-risk subjects, we developed and refined our targeted and multiplexed proteomic test to quantitate only those proteins that were readily and reliably detectable from the initial targeted proteomic assay (n = 32). We then analysed an additional set of 146 longitudinal samples from an independent cohort of 54 individuals at high-risk with iRBD. This cohort was available from the same centre and consisted of longitudinally tracked iRBD subjects who were deeply characterized by PSG and showed additional hyposmia in 88.9% (48/54) of the individuals and additionally α -synuclein Seed Amplification Assay (SAA) positivity in cerebrospinal fluid (CSF) in 91.7 % (22/24) of the subjects as published.¹ Longitudinal follow-up was available for up to 10 years, during which 16 subjects converted to either PD (n = 11) or dementia with Lewy Bodies (DLB; n = 5), jointly summarised as neuronal-synuclein-disorders (NSD). Only serum samples were available in phase II (further details can be found in **Supplementary Table S4**). We investigated how the proteins in our assay correlated between plasma, serum and CSF and found good correlations between plasma and serum, but poor correlations between these matrices and CSF. The limited correlations between blood and CSF proteins correspond to those of other studies comparing the protein expression*

between plasma/serum and CSF^{2,3} and underscore that our test does not necessarily reflect a prodromal and PD-specific proteomic signature of the protein expression in the brain, but rather a potential earlier change in the blood protein expression between healthy controls and very early PD patients. Details from the comparison can be found in **Supplementary Information S1**.

We applied all the longitudinal iRBD samples ($n = 146$) from phase II to the two machine learning models (OPLS-DA and support vector machine) constructed in phase I (PD vs. controls). The OPLS-DA model, based on all 32 detected proteins, identified 70% of the iRBD samples as PD, while the SVM model, which was based on a panel of eight proteins, identified 79% of the samples as PD. At the time of analysis, 16 out of the 54 subjects in our longitudinal iRBD validation cohort had developed an NSD. The earliest correct classification was 7.3 years prior to diagnosis and the latest was 0.9 years prior to diagnosis (average 3.5 ± 2.4 years). Detailed information can be found in **Supplementary Figure S7** and **Supplementary Information S3**.”

Page 2, line 48-68, Supplementary Material: Supplementary Information S3. Evaluation of the larger iRBD cohort, with longitudinal samples (Phase II)

“The longitudinal iRBD samples (146, phase II) were applied to (I) the PD vs. control OPLS-DA model which had been constructed using the initial sample set and (II) our refined and simplified discriminant support vector machine model consisting of eight proteins. The OPLS-DA model (I) contained the detectable proteins and resulted in 107 out of 146 iRBD samples being identified as PD or control, and 39 samples as unclassified. Analysis of this new iRBD samples demonstrated that 70% were consistent identified as having a PD panel profile. Longitudinally, 22 out of the 40 individuals with longitudinal follow-up samples were consistently classified as PD. (II) the SVM model classified 79% of the samples as PD. At the time of analysis, 16 out of the 54 subjects in our longitudinal iRBD validation cohort had developed an NSD. Out of these samples with a known clinical outcome, our SVM model identified ten individuals with all their timepoints as PD. Of the 11 PD converters, 8 were identified as PD. The earliest correct classification was 7.3 years prior to diagnosis and the latest was 0.9 years prior to diagnosis (average 3.5 ± 2.4 years). In 26 of the 40 individuals with longitudinal follow up samples the baseline samples and all longitudinal samples demonstrated a PD profile, while four individuals had all their timepoints classified as control. Ten individuals demonstrated indeterminate predictions, with timepoints from the same individual showing a PD profile in most, but not all samples. Generally, the samples predicted as controls in the SVM model corresponded to the samples with no prediction outcome in the OPLS-DA model. In those subjects that converted to NSD during follow-up, eight of eleven were observed to have the PD profile though all timepoints and the remaining three patients demonstrated a change to a PD profile during follow-up. (**Supplementary Figure S6**).“

Supplementary Figure S6. Classification metrics of the discriminant SVM model, predicting samples as PD or control (phase I)

The classification metrics were calculated from stratified *k*-fold cross validation utilising six splits of the data and 40 repetitions and are displayed as histograms showing the frequency of the metrics precision, recall, the F1 score, the Matthews correlation coefficient and the balanced accuracy score.

Supplementary Figure S7. Prediction results from of a newly acquired set of prodromal iRBD samples (phase II)

146 new serum samples from individuals diagnosed with iRBD, several with longitudinal follow up samples, were predicted in the OPLS-DA model. 70% of the samples were predicted as PD, and 23 of 40 individuals had all their longitudinal samples predicted as PD. In the more refined SVM model, 79% of the 146 new samples were predicted as PD and 27 of 40 individuals consistently had all their longitudinal samples predicted as PD.

Supplementary Table S4. Characteristics of longitudinal iRBD subjects (phase II).

The table shows age, sex, number of longitudinal samples, and the time since baseline for the last sample. Out of the 40 iRBD subjects, 16 had phenoconverted to a neuronal synuclein disease at the time of the last sample (11 Parkinson's disease, 5 Dementia with Lewy bodies). The time from baseline to conversion is shown.

Subject	Age at baseline (years)	Sex	Number of longitudinal Follow-up samples	Time since baseline for final sample (years)	Phenoconversion	Diagnosis
iRBD01	65	Male	5	11	8 years after BL	PD
iRBD02	52	Female	5	10	6 years after BL	PD
iRBD03	77	Female	5	10	8 years after BL	PD
iRBD04	64	Male	4	10		
iRBD05	66	Male	4	10	10 years after BL	DLB
iRBD06	71	Male	3	10		
iRBD07	71	Male	5	10	10 years after BL	PD
iRBD08	62	Male	4	10		
iRBD09	64	Female	5	10	10 years after BL	PD
iRBD10	73	Female	5	10		
iRBD11	69	Male	5	9	9 years after BL	DLB
iRBD12	75	Female	4	9		
iRBD13	73	Male	4	9	9 years after BL	PD
iRBD14	68	Female	3	9		
iRBD15	62	Female	4	9		
iRBD16	64	Male	3	9	9 years after BL	DLB
iRBD17	51	Female	4	8		
iRBD18	63	Male	2	8		
iRBD19	54	Male	3	8		
iRBD20	50	Male	3	8		
iRBD21	77	Male	3	8	8 years after BL	PD
iRBD22	68	Male	4	8	11 mos after BL	DLB
iRBD23	73	Male	2	8		

Subject	Age at baseline (years)	Sex	Number of longitudinal Follow-up samples	Time since baseline for final sample (years)	Phenoconversion	Diagnosis
iRBD24	55	Male	3	7		
iRBD25	53	Male	3	7		
iRBD26	68	Male	3	7	4 years after BL	DLB
iRBD27	72	Female	3	7		
iRBD28	69	Male	2	6	2 years after BL	PD
iRBD29	73	Male	2	6	1 year after BL	PD
iRBD30	77	Male	3	6		
iRBD31	68	Female	2	5		
iRBD32	67	Male	3	5	7 mos after BL	PD
iRBD33	74	Female	3	5		
iRBD34	80	Male	3	5		
iRBD35	58	Male	2	5	1 year after BL	PD
iRBD36	76	Female	2	5		
iRBD37	79	Male	2	4		
iRBD38	70	Male	2	4		
iRBD39	74	Female	2	4		
iRBD40	76	Male	2	4		
Average ± SD	67,5 (8,1)	27 Male	3,3 (1,1)	7,6 (2,1)	3,7 (2,6)	

Abbreviations: iRBD= isolated REM-sleep behavior disorder; PD= Parkinson's disease; DLB= Dementia with Lewy bodies; BL= Baseline; SD= Standard deviation

Reviewer comment: 3. In the current manuscript, all discussions to identify PD patients were conducted by “plasma biomarkers”. I agree with that the establishing plasma biomarker panel is the ultimate goal with considering the clinical accessibility, however the investigations of the “CSF and brain tissues” (hopefully, paired with the plasma samples used for the biomarker panel construction) are also quite important to understand the mechanism of action for the plasma biomarker panel and increasing the reliability. Parkinson’s disease is the CNS-disease, and the alpha-synuclein pathology as well as the damage on dopaminergic neurons are observed in brain parenchyma, not peripheral blood. Here, authors need to add the data and major discussion about the mechanism link between the peripheral biomarker changes and CNS-PD pathology and neurodegeneration. To do that, it is strongly recommended to add the data on CSF and brain tissues (preferably, both the positive and negative control brain regions) proteomics data using the same discovery non-biased method approach.

Our answer: We fully agree that PD is a disease with aggregated aSyn pathology in the CNS. With our study we want to provide insights into proteins in peripheral blood as biomarkers. We indeed have a brain donation program, but so far brain samples of any of these patients donating samples in vivo are not available. With continuing follow-up of these subjects, we will in the future be able to run combined fluid and brain tissue analyses. To at least get closer to the brain, we added cerebrospinal fluid samples, which are most proximal to the brain. Our main goal was to identify a protein pattern in the most peripheral accessible fluid: blood. Knowing that some or most PD starts in the periphery and that there is a lot of peripheral pathology. We are also conducting population based cohort screens for risk factors for PD for the recruitment of upcoming prevention trials.

It is already known that the CSF and blood proteome vary significantly (Whelan, Christopher D., et al. 2019; Acta neuropathologica communications, Dayon et al., 2019; J Proteome Res. We therefore also added these references to the manuscript. Based on basic, fundamental work of Felgenhauer and Reiber, 80% of CSF is a product of blood filtration and only around 20 % of the CSF proteins are brain derived, which could indicate 80% overlap, but further the hydrodynamic radius of the proteins determines its presence in CSF and not all brain areas contribute to the CSF protein composition in the same amount, which has been quite extensively been studied in our institution many years ago (Reiber, Felgenhauer 1987, *Clinica Chimica acta*⁴, Felgenhauer 1974⁵, *Klin Wochenschr*)

To connect our results with the relevant CNS-pathology in PD, we indicate these connections in the discussion section.

We added the following parts to the manuscript/Supplementary Material:

Page 1, line 8-33, Supplementary Material: Supplementary Information S1. Protein expression similarities and differences between plasma, serum and CSF

Supplementary Information S1. Protein expression similarities and differences between plasma, serum and CSF

“We determined the expression of the blood-derived biomarkers in our panel and compared them to those in CSF to ascertain if they potentially could have originated from the brain. Additionally, to determine if serum or plasma gave identical results as **phase 0** and **I** was carried out in plasma samples but for **phase II** analyses only serum samples were available. We analysed our panel in paired plasma, serum, and CSF from subjects with neurological controls. The analysis revealed large differences between the blood- and CSF-protein expressions in a PCA (results can be found in Supplementary figure S7). An OPLS-DA model comparing CSF and plasma/serum was strongly significant (ANOVA $p = 2.0E-20$, permutations $p \ll 0.001$) and demonstrated that SOD3, PTGDS, CST3, DKK3, FABP5, PRG1, PLD3, APOE and HPX were elevated in CSF while all other proteins, apart from HSPA1L, HSPA5 and VCAM1, were higher in the blood-based plasma and serum samples. CSF demonstrated limited correlations with both plasma and serum for all proteins in the predictive panel, apart from MASP2 which was positively correlated (plasma/serum vs CSF, $\rho = 0.66/0.81$). The blood-derived proteins from our panel demonstrated higher values than the CSF proteins for all but DKK3 and PTGDS, which were roughly 80 and 340 times higher, respectively, than the proteins measured in blood. Individual OPLS-DA models of plasma and serum versus CSF did not differ significantly but reflected the observations from the overall blood- versus brain-based model. Comparing plasma and serum, we found that the levels of the proteins FGA, PGK1, SERPINF2, HSPA5, TUBA4A, GRN, HSPA1L and ADIPOQ differed– all of which were higher in plasma, except for ADIPOQ which was higher in serum. Investigating the expression of the proteins in our predictive SVM panel in detail, we correlated the paired samples from the three matrices with each other. We noted that plasma and serum demonstrated positive correlations (Spearman's ρ 0.75 – 0.93) for the five of the eight proteins: C3, GRN, ICAM1, MASP2 and SERPING1, but not for HSPA5, DKK3 and PTGDS, see **Supplementary Figure S8**. Although a high degree of correlation was observed between plasma and serum, the test is considered to perform better in plasma as this is the sample matrix it was developed in.”

Supplementary Figure S8. Protein expression in plasma, serum and CSF.

OPLS-DA scores from a model of plasma and serum versus CSF (top left). The model was highly significant with ANOVA $p = 2.0E-20$ and permutations $p < 0.001$. The corresponding loadings (top right) demonstrated that all but three proteins were significantly different between the blood-based plasma and serum, and CSF. Most of the proteins were elevated in plasma/serum, though HPX, PGK1, APOE, PLD3, FABP5, DKK3, CST3, SOD3 and PTGDS were higher in CSF. Box and whisker plots of the paired plasma, serum and CSF samples, annotated with Spearman's rho and p-value significance levels. The correlation demonstrated that five out of eight of the proteins included in the predictive SVM model were significantly correlated between plasma and serum, post Benjamini-Hochberg multiple testing correction with $\alpha = 0.05$. Only MASP2 exhibited significant correlations between CSF and plasma/serum. The whiskers show the minimum and maximum and the boxes show the 25th percentile, the median and the 75th percentile. *** $p < 0.001$, ** $p < 0.01$, * $p < 0.05$, and ns = not significant.

Page 12, Line 501-506, main manuscript:

“The analysis of paired samples of plasma, serum and CSF revealed only a low correlation between the marker concentrations in peripheral and central compartments. This issue is known and has been reported by several groups before^{2,3}. Nevertheless, our study showed that specific protein patterns in plasma and serum samples in not only PD but also prodromal iRBD are capable to differentiate between the groups, correlate with the clinical picture and that this pattern is already available

in early disease stages. In context with the easy accessibility of blood samples, they show ideal biomarker potential."

Reviewer comment: 4. Authors stated the early detection of PD at even prodromal stage - iRBD can be achieved by the established plasma biomarker panel consisted of the proteins relevant to the inflammation mainly. Indeed, some reports suggest that the inflammation may precede the formation of alpha-synuclein pathology, which implies that the pathogenesis of alpha-synuclein aggregates may be driven by inflammatory pathway. However, this hypothesis is still under discussion, and the authors need to clarify if the developed plasma biomarker panel is proximally relevant to the severity of alpha-synuclein pathology, or just relevant to the PD clinical symptom as clinical phenotype. To clarify this point, one suggested option is that the analysis of pathologically-confirmed (alpha-synucleinopathy with different Braak/Spreading stages) plasma samples analysis, not just the clinically-diagnosed plasma samples. As another option, it is recommended to analyze the PD-model mice - e.g., Line61 etc. with progressing alpha-synuclein pathology in time-course. Even non-clinical studies are also helpful to characterize/understand the mode-of action for the established clinical biomarkers – as a traditional approach.

Our answer: We thank the reviewer for this important point. The manuscript is not designed to confirm the current hypothesis that inflammation precedes α -synuclein aggregation. Although an extremely interesting implication that adds weight to this hypothesis, this would take significant research and is outside the scope of what this manuscript represents or intends to convey. Further, our panel is not only based on proteins, reflecting inflammation but also includes markers of the Wnt-signaling pathway, protein misfolding, ER-stress and neuroprotective mechanisms.

However, the reviewer is correct: we see more inflammation in the prodromal state. This would be an interesting avenue of future research but would require significant funding and time to complete. Furthermore, we believe there are several other groups already working on this hypothesis, our findings may provide significant information to the field to answer this hypothesis and guide researchers attempting to deduce the mechanisms involved in PD.

We focus on analyzing fluid markers in humans, as it is in our opinion the best way to access the natural history of the disease. To our knowledge, there are no animal models available, that reflect the PD specific pathology of the prodromal and clinical stage, including phenoconversion to disease. Animal models have strong limitations, these include the fact that they are based on either toxic (eg 6-OHDA) or genetic (eg A53T α -synuclein -mouse model) mechanisms, which most likely do not reflect the pathology of idiopathic Parkinson's syndrome very well.

Reviewer comment: 5. In Line 195-, authors mentioned, “We additionally predicted the OND samples, out of which nine were predicted as HC, 12 as PD, and 19 were not predicted as belonging to either group. The 12 samples predicted as PD did not demonstrate enrichment in any of the clinical OND groups. The random distribution of the OND samples between PD and HC indicates that the heterogeneous group of OND individuals does not share a distinct protein expression with either the HC or PD groups.” Here, it is necessary to further discuss about the specificity of the developed plasma assay panel for PD (or alpha-synucleinopathy?). For instance, 12 OND participants were classified as PD by the developed plasma biomarker panel, then if so, are there some specific characteristics in these 12 OND participants – e.g., the DLB participants with alpha-synuclein aggregates are enriched in this specific group identified as PD by the developed biomarker panel?

Our answer: The OND subjects represent a very heterogeneous group, we reported the details **in Supplementary Table S1**. The analysis showed no common protein panel between the subjects that were predicted as PD and the clinical OND group. Further, there was no noticeable connection between the diagnosis of the OND subjects and the corresponding protein pattern. We selected the OND group with diagnoses that included differentially diagnostically relevant diagnoses to PD, but due to the heterogeneity of the group and the random distribution, a further subgrouping seemed not to be feasible. To extend the analysis and validate the predictive potential of our model, we added data from the newly acquired set of prodromal iRBD subjects (leading to overall 54 iRBD subjects), consisting of 40 subjects with polysomnography confirmed isolated REM-sleep behavior disorder and available longitudinal serum samples of clinical follow up visits, in some cases up to 10 years. The model consistently predicted most iRBD subjects as PD over time. With the now expanded dataset, we think, that the high number of iRBD subjects predicted as PD points out the specificity of this applied targeted panel of prodromal and early PD. None of the iRBD subjects converted to Multiple Systems Atrophy and only five of them to dementia with Lewy bodies, while 11 converted to PD. One subject showed the PD panel of proteins 7 years before phenoconversion to PD. Therefore, our analysis mostly reports predictive potential of the proteome pattern of the phenoconversion to PD or neuronal α -synuclein disorder (NSD), as it's described in the new publications, introducing a biological staging system for PD⁶ (Simuni et al., 2023; <https://doi.org/10.5281/zenodo.10001310>). We added the following parts to the manuscript and the supplementary material section:

Page 2, line 48-68, Supplementary Material: Supplementary Information S3. Evaluation of the larger iRBD cohort, with longitudinal samples (Phase II)”

“The longitudinal iRBD samples (146, phase II) were applied to (I) the PD vs. control OPLS-DA model which had been constructed using the initial sample set and (II) our refined and simplified discriminant support vector machine model consisting of eight proteins. The OPLS-DA model (I) contained the detectable proteins and resulted in 107 out of 146 iRBD samples being identified as PD or control, and 39 samples as unclassified. Analysis of this new iRBD samples demonstrated that 70% were consistent identified as having a PD panel profile. Longitudinally, 22 out of the 40 individuals with longitudinal follow-up samples were consistently classified as PD. (II) the SVM model classified 79% of the samples as PD. At the time of analysis, 16 out of the 54 subjects in our longitudinal iRBD validation cohort had developed an NSD. Out of these samples with a known clinical outcome, our SVM model identified ten individuals with all their timepoints as PD. Of the 11 PD converters, 8 were identified as PD. The earliest correct classification was 7.3 years prior to diagnosis and the latest was 0.9 years prior to diagnosis (average 3.5 ± 2.4 years). In 26 of the 40 individuals with longitudinal follow up samples the baseline samples and all longitudinal samples demonstrated a PD profile, while four individuals had all their timepoints classified as control. Ten individuals demonstrated indeterminate predictions, with timepoints from the same individual showing a PD profile in most, but not all samples. Generally, the samples predicted as controls in the SVM model corresponded to the samples with no prediction outcome in the OPLS-DA model. In those subjects that converted to NSD during follow-up, eight of eleven were observed to have the PD profile though all timepoints and the remaining three patients demonstrated a change to a PD profile during follow-up (**Supplementary Figure S6**).”

Page 7, line 276-283, main manuscript “We applied all the longitudinal iRBD samples ($n = 146$) from phase II to the two machine learning models (OPLS-DA and support vector machine) constructed in phase I (PD vs. controls). The OPLS-DA model, based on all 32 detected proteins, identified 70% of the iRBD samples as PD, while the SVM model, which was based on a panel of eight proteins, identified 79% of the samples as PD. At the time of analysis, 16 out of the 54 subjects in our longitudinal iRBD validation cohort had developed an NSD. The earliest correct classification was 7.3 years prior to diagnosis and the latest was 0.9 years prior to diagnosis (average 3.5 ± 2.4 years). Detailed information can be found in **Supplementary Figure S7** and **Supplementary Information S3**. “

Page 12, Line 519-537, main manuscript: “Our work was predominantly focused on the similarities between PD and iRBD, particularly as PD is the most common clinical syndrome developing out of iRBD. Previous proteomic analysis has been able to distinguish PD from DLB⁷, but data on DLB and iRBD is lacking. Future work would include refinement of the panels of biomarkers developed in this

study and using the pipeline described in this manuscript, to identify and validate additional biomarkers that could distinguish between iRBD and PD, MSA and DLB. Another advantage of using triple quadrupole platforms is that new and better biomarkers can easily be augmented into the test described in this manuscript. Thus, any test could be refined and optimised over time with very little modification to the assay as new biomarkers are discovered. In summary, instead of single biomarkers, in a univariate approach, we have created a pipeline using a targeted proteomic test of a multiplexed panel of proteins, together with machine learning. This powerful combination of multiple well-selected biomarkers with state-of-the-art machine learning bioinformatics, allowed us to use a panel of eight biomarkers which enabled us to distinguish early PD from HC. This biomarker panel provided a distinct signature of protective and detrimental mechanisms, finally triggering oxidative stress and neuroinflammation, leading to α -synuclein aggregation and LB formation. Moreover, this signature was already present in the prodromal stages of the disease, before motor onset, supporting the high specificity of iRBD and its high conversion rate to a NSD especially PD⁸. And most important this blood panel can in the future help to identify subjects at risk to develop neuronal synuclein aggregation disorder and stratify subjects for upcoming prevention trials."

Supplementary Figure S6. Classification metrics of the discriminant SVM model, predicting samples as PD or control (phase I)

The classification metrics were calculated from stratified k-fold cross validation utilising six splits of the data and 40 repetitions and are displayed as histograms showing the frequency of the metrics precision, recall, the F1 score, the Matthews correlation coefficient and the balanced accuracy score.

Supplementary Figure S7. Prediction results from of a newly acquired set of prodromal iRBD samples (phase II)

146 new serum samples from individuals diagnosed with iRBD, several with longitudinal follow up samples, were predicted in the OPLS-DA model. 70% of the samples were predicted as PD, and 23 of 40 individuals had all their longitudinal samples predicted as PD. In the more refined SVM model, 79% of the 146 new samples were predicted as PD and 27 of 40 individuals consistently had all their longitudinal samples predicted as PD.

Reviewer comment: 6. In the current manuscript, the bidirectional performance to diagnose e.g., HC vs PD by the developed biomarker panel is mainly discussed. Here, authors should investigate if the developed panel can be applied to assess the disease severity in terms of the clinical scales and also the amount of pathology – i.e., alpha-synuclein aggregates in brain. In Supplementary Table 2, some data for Spearman correlations between each protein biomarker and clinical scales are described, however it seems that all individual rho values are very low and difficult to be used to predict the severity in a quantitative manner. Even though the individual proteins do not show the convincing results on predicting the disease severity, some regression models with composite score can be constructed for the quantitative assessment, not the bidirectional diagnosis purpose only?

Our answer: We thank the reviewer for the comment and agree that the combination of clinical scores and the brain pathology could provide new insights into the disease specific pathology and its correlation with the clinical picture. The authors have applied for funding to carry out this because it would require significant funds, a unique collection of samples that would need to be collected and time to address this. We believe this would be part of a follow up mechanism paper. We also have a brain-donation program and await together with further longitudinal follow-up, increase number of converters and

eventually the collection of more brain material in the upcoming years for more analyses. Therefore, to answer this question pondered by the reviewer may take another 20-30 years before such samples are even available. Thus, we believe it is not possible to answer the reviewer's point in the timeframe available to us.

With the additional longitudinal data sets that we added to the MS we were also able to extend the correlation analysis and included a linear mixed effects model. The results can be found in **Supplementary Table S6**.

We decided not to work with composite scores, as the main aim of the study was the discriminatory potential of the proteome pattern and not the assessment of clinical phenotypes.

We added the following part to the main manuscript:

Page: 7, line: 307-326 main manuscript: Comparison with clinical outcomes and measurements in the longitudinal iRBD samples (phase II)

*“The longitudinal expression in the iRBD samples was evaluated using linear mixed effects models. Conditional growth models with random slope and random intercept between the individuals were constructed. After adjusting the p-values for multiple testing by applying the Benjamini-Hochberg (BH) procedure with alpha = 0.05, we found that BCHE was significantly decreased over the timepoints in the iRBD individuals ($p = 0.01$). We next focussed on only the iRBD samples with at least two timepoints and which had consistently been predicted as PD in the SVM model ($n = 90$). This produced comparable results to the initial model with BCHE significantly related with time since baseline ($p = 0.01$), but also TUBA4A nominally significant ($p = 0.04$) although not passing the BH FDR threshold. The modelling also demonstrated that the clinical measurements Hoehn-Yahr ($p = 0.02$), UPDRS I-III ($p = 0.02$), and UPDRS I and III ($p = 0.03$ and 0.03 , respectively), were significantly related with the time since baseline in the iRBD group post multiple testing correction. The summed PD non-motor symptoms measurement was strongly correlated with longitudinal progression ($p = 5E^{-8}$), as were the questionnaire PDQ-39's mean values ($p = 0.005$). From routine blood values available, Cholesterol was moreover associated with the longitudinal timepoints ($p = 0.02$). Details can be found in **Supplementary Table S6**. Correlating the clinical measurements with the targeted proteomic data, we applied Spearman correlation and found that cholesterol was positively correlated with six proteins (**Supplementary Table S7**) including HSPA8, APOE and MASP2 ($p = 5E^{-9}$, 0.0003 and 0.003 , respectively). Also correlated, but to a lesser degree and not passing the BH FDR-threshold were the PD NMS sum which correlated negatively with TUBA4A (p unadjusted = 0.01) and the PDQ-39 mean values which correlated negatively with CST3 and PTGDS (p unadjusted = 0.03 and 0.05 , respectively).”*

Supplementary Table S6: Supplementary Material: Significant relationships with longitudinal progression evaluated by linear mixed effects models (phase II).

The table shows the p-value significance of the interaction between the time since baseline and each significant clinical variable post Benjamini-Hochberg multiple testing correction (alpha = 0.05), the coefficient and the 95% confidence interval \pm the standard error. The p-values are denoted by **** $p < 0.0001$, *** $p < 0.001$, ** $p < 0.01$, and * $p < 0.05$.

Clinical variable	Significance	Coefficient	95 CI \pm SE
Hoehn-Yahr	*	0.011	[0.0038, 0.0181] \pm 0.0036
UPDRS I	*	0.018	[0.0041, 0.0311] \pm 0.0069
UPDRS III	*	0.098	[0.0252, 0.1698] \pm 0.0369
UPDRS I-III	*	0.16	[0.0547, 0.2608] \pm 0.0526
UPDRS (sum)	*	0.16	[0.0549, 0.261] \pm 0.0526
PD non-motor symptoms measurement (sum)	****	0.12	[0.0938, 0.1474] \pm 0.0137
PD non-motor symptoms measurement (mean)	**	0.0035	[0.0018, 0.0053] \pm 0.0009
Cholesterol	*	-0.30	[-0.4841, -0.1186] \pm 0.0932

Abbreviations: MMSE= Mini-Mental State Examination, UPDRS= Unified Parkinson's Disease Rating Scale)

Supplementary Table S7: Supplementary Material: Significant correlations between cholesterol and proteins measured by targeted mass spectrometry, evaluated by Spearman correlation (phase II).

The table shows the p-value significance of the correlations between cholesterol and the significant proteins post Benjamini-Hochberg multiple testing correction (alpha = 0.05). the correlation coefficient is also shown. The p-values are denoted by **** $p < 0.0001$, *** $p < 0.001$, ** $p < 0.01$, and * $p < 0.05$.

Protein	Significance	Correlation coefficient
HSPA8	****	0.50
APOE	***	0.45
MASP2	**	0.39
PRG4	*	0.35
BCHE	*	0.31
SERPINA1	*	0.31

Reviewer #2 (Remarks to the Author):

The authors sought to investigate whether a panel of plasma proteomic biomarkers could discriminate PD from healthy controls, and also be applicable to iRBD.

There are many positive aspects of this analysis and manuscript, which the authors describe very well. The manuscript is also well-written, and the figures and tables are informative.

Please address the following points:

Reviewer comment: The authors state that “Parkinson’s disease (PD) is an increasingly prevalent neurodegenerative disease” – please provide references that PD is increasing in prevalence.

Our answer: We added the following references to the manuscript:

Page 8, line 329-336, main manuscript: “PD has emerged as the world’s fastest growing neurodegenerative disorder and is currently affecting close to 10 million people worldwide, thereby emphasizing the urgent need for disease-modifying and prevention strategies^{9,10}”.

Bloem, B.R., Okun, M.S. & Klein, C. Parkinson's disease. Lancet 397, 2284-2303 (2021).

Dorsey, E.R., Sherer, T., Okun, M.S. & Bloem, B.R. The Emerging Evidence of the Parkinson Pandemic. J Parkinsons Dis 8, S3-s8 (2018).

Reviewer comment: Note the error in lines 142-143 – “Further details can be found in Supplementary Error! Reference source not found”

Our answer: We apologize for this error and have corrected it.

Reviewer comment: In the discussion, considering the recent associations of GRN mutations, lysosomal function, Lewy body dementia, etc. (for example, see Reho et al, Mov Disord 2022), it would be worth adding comments and associated references on this topic.

Our answer: We thank the reviewer for this important comment as GRN dysfunction is actually widely discussed in the field. We added a reference and a part to the discussion section:

Page 10/11, line 443-452, main manuscript: *"In addition, a strong downregulation of progranulin (GRN) was detected, indicating a potential loss of neuroprotection and increased susceptibility to neuroinflammation. GRN may act as a neurotrophic factor, promoting neuronal survival and modulating lysosomal function. Loss-of-function mutations in the GRN gene are a cause of frontotemporal dementia and also of familial Dementia with Lewy bodies. GRN gene variants are also known to increase the risk of developing AD and PD¹¹. The main characteristics of neurodegeneration related to GRN are TDP43-inclusions, but LB-pathology is also very common. Loss of progranulin has further been linked to increased production of pro-inflammatory species such as TNF and IL-6 in microglia¹². A study in mice showed that Grn^{-/-} mice had elevated levels of complement proteins, including C3 even before the onset of neurodegeneration¹³. Additionally, previous studies have found GRN downregulated in serum samples of advanced PD compared to Alzheimer's disease and healthy individuals¹⁴."*

Reviewer comment: This analysis obviously focused on proteomic plasma biomarkers in PD. The authors should comment on the findings as they related to dementia with Lewy bodies (DLB) and MSA. Comments would be particularly needed to explain the findings in the small iRBD cohort. The panel predicted 72-94% of the iRBD samples as PD, which could be due to several factors/issues. One could be the possible change in the natural evolution of proteins from the prodromal (eg, iRBD) to full PD clinical syndrome phase, which the authors allude to. Another explanation is that some of the iRBD patients will evolve into a full DLB or MSA?? clinical syndromes. The proteomic panel may or may not be different between PD and DLB, and the authors should compare and contrast their findings with those of O'Bryant et al, Alz and Dem 2019. While one might predict that the panel would be different between NSD and glial synuclein aggregation disorder (i.e. MSA), that is also an open question, which should also be discussed when more samples with longitudinal follow-up and phenoconversion will be available.

Our answer: This is an excellent point from the reviewer. Therefore, we have added this suggestion to the discussion including the relevance to the findings of Bryant et al.

Our study did not test the differences and similarities of the proteome pattern of MSA subjects. As only four iRBD subjects converted to DLB, we also have no extended data on this. We strongly focused on iRBD and PD/DLB as the most common neuronal α -synuclein aggregation disorders.

We added the following parts to the manuscript:

Page 12, Line 519-537, main manuscript: *“Our work was predominantly focused on the similarities between PD and iRBD, particularly as PD is the most common clinical syndrome developing out of iRBD. Previous proteomic analysis has been able to distinguish PD from DLB⁷, but data on DLB and iRBD is lacking. Future work would include refinement of the panels of biomarkers developed in this study and using the pipeline described in this manuscript, to identify and validate additional biomarkers that could distinguish between iRBD and PD, MSA and DLB. Another advantage of using triple quadrupole platforms is that new and better biomarkers can easily be augmented into the test described in this manuscript. Thus, any test could be refined and optimised over time with very little modification to the assay as new biomarkers are discovered. In summary, instead of single biomarkers, in a univariate approach, we have created a pipeline using a targeted proteomic test of a multiplexed panel of proteins, together with machine learning. This powerful combination of multiple well-selected biomarkers with state-of-the-art machine learning bioinformatics, allowed us to use a panel of eight biomarkers which enabled us to distinguish early PD from HC. This biomarker panel provided a distinct signature of protective and detrimental mechanisms, finally triggering oxidative stress and neuroinflammation, leading to α -synuclein aggregation and LB formation. Moreover, this signature was already present in the prodromal stages of the disease, before motor onset, supporting the high specificity of iRBD and its high conversion rate to a NSD especially PD⁸. And most important this blood panel can in the future help to identify subjects at risk to develop neuronal synuclein aggregation disorder and stratify subjects for upcoming prevention trials.”*

Reviewer #3 (Remarks to the Author):

Summary

In their study, Hällqvist et. al. present a panel of proteomic plasma biomarkers that are capable of discriminating PD and healthy controls with perfect accuracy in a machine learning model. They apply their model on prodromal subjects with iRBD and predict samples with a similar rate as expected from clinical conversion rates. Overall, the study is well-written and concise. The figures are informative. Mass spectrometry based plasma proteomics to uncover biomarker panels has been studied for several years now. Although cerebrospinal fluid (CSF) is often the preferred body fluid for Parkinson's research, plasma is preferred due to its potential for clinical applications, owing to its ease of access.

Reviewer comment: In current studies, a machine learning (ML) layer is often added following the identification of proteins of interest. This layer serves to build predictive models, a process

that is becoming increasingly standardized. Sometimes, these studies correlate clinical parameters with protein values to provide deeper insights. Novel studies in this field usually leverage recent technological advancements to achieve greater protein depth, involving larger cohorts for increased reliability, or utilizing innovative algorithms for data interpretation. An early example reference would be Pan et al. (Journal of Proteome Research 2014, <https://doi.org/10.1021/pr500421v>), which used targeted plasma proteomics to study biomarkers generated from CSF measurements on a cohort of 282 patients, achieving an AUC of 0.753 with four peptides with a linear model. In evaluating the current study, however, there seems to be a lack of distinctive technological advancement, an unusually large cohort, or an innovative algorithm. There doesn't appear to be any significant leap from the existing state-of-the-art, based on the manuscript text. Yet, this should not overshadow the development of a protein panel that exhibits high predictive power and can achieve impeccable accuracy. Such an achievement could indeed be regarded as a form of novelty that is of great interest for the community.

Our answer: We thank the reviewer for this comment. With this study we aim to translate an easy to apply targeted mass spec panel, that enables the differentiation between HC and PD and iRBD, and is capable to successfully correlate the proteomic markers to the clinical picture of the subjects. We used the ML models as useful tools to optimize and validate our model. The development of new AI algorithms and MS approaches is an important subject but not the main focus of our manuscript. As a new feature, we can provide for the first time longitudinal proteome data from iRBD subjects and developed a rapid and refined LC-MS/MS method, possible to run in any laboratory with an available tandem LC-MS instrument.

We added the following part to the manuscript:

Page 8/9, line 343-360, main manuscript: *"Other emerging multiplex technologies are increasingly used to identify individual proteomic biomarkers. However, these techniques are not true proteomic or 'eyes open' methods, as they rely on selected large panels of specific antibodies/and other (e.g., aptamer)-based assay technologies. These techniques, although useful, have not provided consistent results^{15,16}. In our own research we applied these techniques in our PD cohort^{17,18}, and identified several pathways including inflammation that were perturbed, but we were not able to validate the single proteins in an unbiased manner. Proteomics using mass spectrometry measures all expressed proteins in an unbiased fashion as opposed to those selectively included in a panel. Therefore, proteomic screening using mass spectrometry-based techniques is much more likely to identify new pathways or biomarkers and provides more meaningful insights into the disease mechanisms involved in PD. We found a discrepancy between the detected markers between the discovery and the targeted phase. This is a known phenomenon in biomarker translation¹⁹, reflected in the low number of biomarkers that got an FDA approval²⁰.*

We addressed this by refining our panel, reducing the number of markers and increasing the sample size, what has been reported as successful improvement strategies²¹. Further, the validation of potential biomarkers was performed on a second and different type of mass spectrometer (triple quadrupole), which added the advantage of being available in all large hospitals. Thus, a significant strength of our biomarker discovery to translation pipeline is that it allows for the developed test to be easily validated and translated to any clinical laboratory equipped with a tandem LC-MS instrument."

Page 9, line 362-365, main manuscript: "Targeted MS has been applied in PD previously, including by the authors, but typically the biological fluid used has been CSF²² and not with readily available peripheral fluids such as the low amount of plasma/serum required in our study (10 µl). In addition, the authors are unaware of any study that have analysed longitudinal studies and prodromal cohorts including iRBD and phenoconverters."

Reviewer comment: Major: Study Design

For the above reasons, I find it extremely important how this predictive panel was generated. The authors conduct a proteomics discovery study and identify 47 proteins as differentially expressed. However, their final multiplexed assay consists of 121 proteins, stemming from "unpublished discovery studies", AD, and proteins in the literature. The referenced literature includes an Alzheimer's mouse model from 2005 and a study of Neuroinflammation in Schizophrenia. Essentially, there is a well-defined statistical framework for 47 proteins, and then 74 more are added in an opaque way. This, however, invalidates the whole discovery approach and severely limits the biological significance of the subsequent targeted analysis. To address this, I would recommend the following:

- State how the final list of proteins was generated and included a list of them. In line 502, it is referenced to be Supplementary Table 2, but this is the correlation to clinical parameters. It is evident that the community will derive little value if the highly predictive panel is not disclosed.
- Conduct a comparison between the significant proteins from the discovery phase and the proteins featured in the targeted assay. There seems to be a discrepancy, as some of the highly regulated proteins from Supplementary Figure S1 are not present in Figure 1. What is the authors' explanation for this? Maintaining consistency in the representation of data, such as using either lollipop charts or volcano plots (or ideally both), would facilitate interpretation.

Our answer: We thank the reviewer for these important points in regard of the applied protein panel.

We have now clarified that several proteins in the targeted panel originated from our in-house neuroinflammatory panel, which was developed from

literature and several discovery studies, one of which we have since published²³. We further cited two publications where the targeted neuroinflammatory panel has been applied^{24,25}. Further, we have expanded in the introduction and discussion why the researchers created the final targeted MRM panel. Additionally, all protein names, MRMs and experimental parameters are included in the manuscript supplementary information so any researcher can reproduce or use the predictive panel (**Supplementary Table S3**). By providing information on the proteins including the gene ID, protein name, amino acid sequence and MRMs, we increased transparency and traceability of the applied techniques.

We found a degree of discrepancy between the results from the discovery and the targeted phases. This was not unexpected as it is well-known that the translation of proteomic biomarkers, going from discovery, to validation and finally to clinical translation, is challenging¹⁹ which is clearly conveyed by the low number of proteomic biomarkers currently approved by the Food and Drug Administration (FDA)²⁰. The reviewer is correct, and it is our fault for not explaining this common issue in medical proteomics and translation of biomarkers into clinical practice more clearly. It is common practice in clinical proteomics that all omics results are followed up by a secondary and different validity test and usually using another technique. This is why so many biomarkers fail to translate to clinical practice and why we have developed the pipeline described in this paper i.e. validation of potential biomarkers is performed on a second and different type of mass spectrometer. In addition, this second type of mass spectrometer (triple quadrupole) which was used in this study has the added advantage of being available in all large hospitals. The main aim and advantage of this work was the creation of a test which can be performed in most large clinical chemistry laboratories for added patient benefit. Reducing the number of markers can be an effective tool to optimize the translational potential, so we refined our mass-spec panel and increased the sample size to increase the detectability of the proteins and facilitate the possible translation of the panel into clinical diagnostics²¹. However, we find that a large number of the proteins included in the targeted assay translated successfully between phase I and phase II as demonstrated by the machine learning predictions.

To address these points and make it clearer, we have added the following lines and additional references to the manuscript:

Page 3, line 104-119, main manuscript: *"In this study, we employed mass spectrometry-based proteomic phenotyping to identify a panel of blood biomarkers in early PD. In the initial discovery stage termed **phase 0**, we analysed samples from a well-characterized de novo PD patients and healthy controls (HC) that had been*

subjected to rigorous collection protocols.²⁶ Using unbiased state-of-the-art mass spectrometry, we identified putatively involved proteins, suggesting an early inflammatory profile in plasma. We thereafter moved on to validation in **phase I**, by creating a high throughput and targeted proteomic assay which was applied to plasma samples from an independent replication cohort, consisting of de novo PD, HC, and iRBD patients. Finally after refining the targeted proteomic panel to include a multiplex of only the biomarkers which were reliably measured, an independent analysis (**phase II**) was performed on a larger and independent, cohort of longitudinal, high-risk subjects who had been confirmed as iRBD by state-of-the art video recorded polysomnography (vPSG), including follow-up sampling of up to 7 years.

In summary, using a panel of eight blood biomarkers identified in a machine learning approach, allowed us (I) to differentiate between PD and HC with a specificity of 100%, and (II) to predict 79% of the iRBD subjects as PD up to 7 years before conversion to motor PD-. Our identified panel of biomarkers significantly advances PD research by providing potential screening and detection markers for use in the earliest stages of PD for subject identification/stratification for the upcoming prevention trials."

Page 8/9, line 343-360, main manuscript: "Other emerging multiplex technologies are increasingly used to identify individual proteomic biomarkers. However, these techniques are not true proteomic or 'eyes open' methods, as they rely on selected large panels of specific antibodies/and other (e.g., aptamer)-based assay technologies. These techniques, although useful, have not provided consistent results^{15,16}. In our own research we applied these techniques in our PD cohort^{17,18}, and identified several pathways including inflammation that were perturbed, but we were not able to validate the single proteins in an unbiased manner. Proteomics using mass spectrometry measures all expressed proteins in an unbiased fashion as opposed to those selectively included in a panel. Therefore, proteomic screening using mass spectrometry-based techniques is much more likely to identify new pathways or biomarkers and provides more meaningful insights into the disease mechanisms involved in PD. We found a discrepancy between the detected markers between the discovery and the targeted phase. This is a known phenomenon in biomarker translation¹⁹, reflected in the low number of biomarkers that got an FDA approval²⁰. We addressed this by refining our panel, reducing the number of markers and increasing the sample size, what has been reported as successful improvement strategies²¹. Further, the validation of potential biomarkers was performed on a second and different type of mass spectrometer (triple quadrupole), which added the advantage of being available in all large hospitals. Thus, a significant strength of our biomarker discovery to translation pipeline is that it allows for the developed test to be easily validated and translated to any clinical laboratory equipped with a tandem LC-MS instrument."

Page 9, line 362-365, main manuscript: "Targeted MS has been applied in PD previously, including by the authors, but typically the biological fluid used has been CSF²² and not with readily available peripheral fluids such as the low amount of

plasma/serum required in our study (10 μ l). In addition, the authors are unaware of any study that have analysed longitudinal studies and prodromal cohorts including iRBD and phenoconverters."

Reviewer comment: - Enhance the characterization of the identified proteins. While a machine learning model provides a quantitative metric of model performance, it would be beneficial to see additional technical controls. These could include correlation plots with hierarchical clustering and box/swarm plots for relevant proteins.

Our answer: This is a very good suggestion and we have now included Box-scatter plots of all the significantly different proteins in **Supplementary Figure S2**. We have moreover added hierarchical clustering to the correlation plot in **Figure 4**. Please see below.

Supplementary Figure S3. Significantly different proteins between controls and the different disease groups DNP, iRBD and OND (phase II).

The data are displayed as Box and whisker plots overlaid with scatter plots of the individual measurements. The whiskers show the minimum and maximum and the boxes show the 25th percentile, the median and the 75th percentile. ns = not significant, * $p < 0.05$, ** $p < 0.01$, *** $p < 0.001$, and **** $p < 0.0001$. The proteins are represented by gene names.

Figure 4. Correlation and clustering heatmap of proteins measured by targeted mass spectrometry and clinical scores in controls and PD subjects. (phase I)

The correlation was performed using Spearman's procedure, and the clustering method was set to average. The clustering metric was Euclidean. The heatmap is coloured by correlation coefficient where red represents positive and blue negative correlations. Detailed information about the protein correlations can be found in Supplementary Table S3. De novo Parkinson's disease ($n = 99$) and healthy controls ($n = 36$). Abbreviations: MMSE= Mini Mental State Examination, UPDRS= Unified Parkinson's Disease Rating Scale

Reviewer comment: Major: ML model

The characterization of the machine learning (ML) model in this study could benefit from more thorough detailing. A receiver operating characteristic (ROC) curve based solely on the training set doesn't provide a comprehensive understanding of the model's generalizability. This analysis should be conducted on the test set. Additionally, the report lacks standard classification metrics such as precision, recall, and F1 score. Given that the dataset is imbalanced, with 99 Parkinson's Disease (PD) cases versus 36 healthy controls, a model that categorically predicts PD would already achieve 73% accuracy. Therefore, balanced accuracy or the Matthews correlation coefficient (MCC) should be considered for a more accurate evaluation. Alongside the ROC curve, a precision-recall (PR) curve would be a valuable additional metric. The section spanning lines 581-585 is somewhat ambiguous. Was the 5-fold split applied to the 70% of the data? Without access to the ML code, it's difficult to fully understand the processes involved. While it's commendable that the feature extraction step was cross-validated, given the limited sample size, it would be very beneficial to also cross-validate the entire pipeline, rather than relying on a single split—ideally repeatedly. Scikit-learn's Repeated KFold could be readily implemented for this purpose.

Our answer: We thank the reviewer for these helpful suggestions. We agree that the mentioned metrics would aid in the interpretability of the model's performance and have since made the following edits:

- ROC curve conducted on training set
- Precision-recall curve performed on the training set and added to Figure 2
- Precision, recall, F1-score, Matthew's correlation coefficient and balanced accuracy score extracted from a stratified k-fold cross validation of the whole dataset partitioning the data into six splits with 100 repetitions each (Supplementary Figure S5)
- The method of cross-validation of the feature selection has been clarified

We added the following text:

Page 6, line 237-243, main manuscript: *"The combined panel was attributed with ROC and PR areas under the curve (AUC) of 1.0, while the values for the individual predictors ranged from 0.53 to 0.92 in the ROC curve, and from 0.79 to 0.96 in the PR curve (Figure 2). We further evaluated the whole dataset by performing repeated cross validation with six splits of the data and 40 repetitions. The resulting classification metrics (Supplementary Figure S6) demonstrated average precision, recall, F1 score, and balanced accuracy score of 0.995, 0.994, 0.995 and 0.987, respectively and thereby indicating a highly robust classification model."*

Supplementary Figure S6. Classification metrics of the discriminant SVM model, predicting samples as PD or control (phase I)

The classification metrics were calculated from stratified k-fold cross validation utilising six splits of the data and 40 repetitions and are displayed as histograms showing the frequency of the metrics precision, recall, the F1 score, the Matthews correlation coefficient and the balanced accuracy score.

Reviewer comment: found the correlation to clinical scores interesting, but, also confusing: On one hand, the aim is to develop a predictive biomarker panel, while on the other, it's desirable to identify biomarkers that correlate with the scale. Wouldn't it make more sense to directly construct an ML model capable of predicting, for example, the clinical rating scale?

Our answer: We thank the reviewer for this comment. We agree that the prediction of clinical rating scales, such as UPDRS or MMSE, could provide valuable insights. In this paper, our main goal was to predict the individuals' diagnoses, although we found the clinical correlations informative and helpful to corroborate the variable selection in the ML model. Since the submission of the original draft, we have included a large new cohort of individuals diagnosed with iRBD (54 subjects), proven in the polysomnography, several with up to 5 longitudinal timepoints and 10 years of follow up, and predicted these in the ML model. We further evaluated the relationship between the clinical variables and the proteins.

Page: 7, line: 307-326, main manuscript: Comparison with clinical outcomes and measurements in the longitudinal iRBD samples (phase II) "The longitudinal expression in the iRBD samples was evaluated using linear mixed effects models. Conditional growth models with random slope and random intercept between the individuals were constructed. After adjusting the p-values for multiple testing by applying the Benjamini-Hochberg (BH) procedure with $\alpha = 0.05$, we found that BCHE was significantly decreased over the timepoints in the iRBD individuals ($p = 0.01$). We next focussed on only the iRBD samples with at least two timepoints and which had consistently been predicted as PD in the SVM model ($n = 90$). This produced comparable results to the initial model with BCHE significantly related with time since baseline ($p = 0.01$), but also TUBA4A nominally significant ($p = 0.04$) although not passing the BH FDR threshold. The modelling also demonstrated that the clinical measurements Hoehn-Yahr ($p = 0.02$), UPDRS I-III ($p = 0.02$), and UPDRS I and III ($p = 0.03$ and 0.03 , respectively), were significantly related with the time since baseline in the

*iRBD group post multiple testing correction. The summed PD non-motor symptoms measurement was strongly correlated with longitudinal progression ($p = 5E^{-8}$), as were the questionnaire PDQ-39's mean values ($p = 0.005$). From routine blood values available, Cholesterol was moreover associated with the longitudinal timepoints ($p = 0.02$). Details can be found in **Supplementary Table S6**. Correlating the clinical measurements with the targeted proteomic data, we applied Spearman correlation and found that cholesterol was positively correlated with six proteins (**Supplementary Table S7**) including HSPA8, APOE and MASP2 ($p = 5E^{-9}$, 0.0003 and 0.003, respectively). Also correlated, but to a lesser degree and not passing the BH FDR-threshold were the PD NMS sum which correlated negatively with TUBA4A (p unadjusted = 0.01) and the PDQ-39 mean values which correlated negatively with CST3 and PTGDS (p unadjusted = 0.03 and 0.05, respectively)."*

Reviewer comment: Protein panel validation

The authors could certainly benefit from providing a more thorough analysis of the significant proteins identified during both the discovery and targeted phases. Presently, there appears to be a degree of inconsistency in terms of the significant proteins identified, and this is somewhat perplexing. A comparison with existing literature could provide valuable insights and potentially explain these discrepancies.

The authors do this partly by comparing pathways, and, e.g., comparing to their previous studies, but when comparing to the volcano plot of their OLINK study there seem to be entirely different proteins. This could be done by providing a table with proteins and references where they were identified previously.

Our answer: We thank the reviewer for mentioning this important aspect. We cited the previous research to set our MS approach into the context of the actual research. As we mentioned above it is a known phenomenon, that proteins, identified in multi-omics approaches are not validated in other cohorts with different methods. The mentioned OLINK panel is a targeted panel based on another technology (proximity extension assay). We did not intent to run a validation or one to one comparisons of the applied methods.

We added the following lines to the manuscript to emphasize this point:

Page 8/9, line 343-360, main manuscript: *"Other emerging multiplex technologies are increasingly used to identify individual proteomic biomarkers. However, these techniques are not true proteomic or 'eyes open' methods, as they rely on selected large panels of specific antibodies/and other (e.g., aptamer)-based assay technologies. These techniques, although useful, have not provided consistent results^{15,16}. In our own research we applied these techniques in our PD cohort^{17,18}, and identified several pathways including inflammation that were perturbed, but we were*

not able to validate the single proteins in an unbiased manner. Proteomics using mass spectrometry measures all expressed proteins in an unbiased fashion as opposed to those selectively included in a panel. Therefore, proteomic screening using mass spectrometry-based techniques is much more likely to identify new pathways or biomarkers and provides more meaningful insights into the disease mechanisms involved in PD. We found a discrepancy between the detected markers between the discovery and the targeted phase. This is a known phenomenon in biomarker translation¹⁹, reflected in the low number of biomarkers that got an FDA approval²⁰. We addressed this by refining our panel, reducing the number of markers and increasing the sample size, what has been reported as successful improvement strategies²¹. Further, the validation of potential biomarkers was performed on a second and different type of mass spectrometer (triple quadrupole), which added the advantage of being available in all large hospitals. Thus, a significant strength of our biomarker discovery to translation pipeline is that it allows for the developed test to be easily validated and translated to any clinical laboratory equipped with a tandem LC-MS instrument."

Reviewer comment:

Misc: - The auto-generated PDF appears to have formatting issues, e.g. Reference source not found, figures are not readable

Our answer: We apologize for any inconvenience caused, rechecked the references and submitted all images in high resolution in the revised version.

Reviewer comment: - Figure 2: The figure contains long descriptions for the Proteins, whereas the text has short names. It would increase readability to stick to one.

Our answer: Thanks for this comments, to increase readability we adapted the caption accordingly.

Reviewer comment: - L398: Clinical testing for neurological disorders is limited to use of a select few well characterised individual markers and translating new biomarkers to eventual clinical application is notoriously challenging. "to use of a"

Our answer: We corrected this.

Page 13, line 540-548, main manuscript: *Clinical testing for neurological disorders is limited to the use of a selected few well characterized individual markers and translating new biomarkers to eventual clinical application is notoriously challenging. The power of using multiplexed biomarker technologies with machine learning enables biomarkers to be evaluated in context with other markers of pathological events thereby creating a 'disease profile' as opposed to individual markers.*

Reviewer comment: In conclusion, I cannot endorse this study for publication in its current form in Nature Communications. However, I believe that a highly predictive biomarker panel,

as presented in this study, could be of considerable interest to the scientific community. Therefore, I would support the publication of this study, provided the necessary revisions are made

Our answer: We thank the reviewer for this comment that if we were able to analyse the extra samples, correlate different tissue types and took on board all suggestions they would support this work for publication. We therefore appreciate the constructive comments of the reviewers and hope that the revisions merit the editors and the reviewers. We believe the manuscript gained significantly with the additional cohort and analyses.

References

- 1 Concha-Marambio, L. *et al.* Accurate Detection of α -Synuclein Seeds in Cerebrospinal Fluid from Isolated Rapid Eye Movement Sleep Behavior Disorder and Patients with Parkinson's Disease in the DeNovo Parkinson (DeNoPa) Cohort. *Mov Disord* **38**, 567-578, doi:10.1002/mds.29329 (2023).
- 2 Dayon, L. *et al.* Proteomes of Paired Human Cerebrospinal Fluid and Plasma: Relation to Blood-Brain Barrier Permeability in Older Adults. *J Proteome Res* **18**, 1162-1174, doi:10.1021/acs.jproteome.8b00809 (2019).
- 3 Whelan, C. D. *et al.* Multiplex proteomics identifies novel CSF and plasma biomarkers of early Alzheimer's disease. *Acta Neuropathol Commun* **7**, 169, doi:10.1186/s40478-019-0795-2 (2019).
- 4 Reiber, H. & Felgenhauer, K. Protein transfer at the blood cerebrospinal fluid barrier and the quantitation of the humoral immune response within the central nervous system. *Clin Chim Acta* **163**, 319-328, doi:10.1016/0009-8981(87)90250-6 (1987).
- 5 Felgenhauer, K. Protein size and cerebrospinal fluid composition. *Klinische Wochenschrift* **52**, 1158-1164, doi:10.1007/BF01466734 (1974).
- 6 Simuni, T., Chahine, L., Poston, K., Brumm, M., Buracchio, T., Campbell, M., Chowdhury, S., Coffey, C., Concha-Marambio, L., Dam, T., DiBioso, P., Foroud, T., Frasier, M., Gochanour, C., Jennings, D., Kiebertz, K., Kopil, C. M., Merchant, K., Mollenhauer, B., ... Marek, K. . Biological Definition of Neuronal alpha-Synuclein Disease: Towards an Integrated Staging System for Research. *Lancet Neurology*, doi: <https://doi.org/10.5281/zenodo.10001310> (2023).
- 7 O'Bryant, S. E. *et al.* A proteomic signature for dementia with Lewy bodies. *Alzheimers Dement (Amst)* **11**, 270-276, doi:10.1016/j.dadm.2019.01.006 (2019).
- 8 Hu, M. T. REM sleep behavior disorder (RBD). *Neurobiol Dis* **143**, 104996, doi:10.1016/j.nbd.2020.104996 (2020).
- 9 Bloem, B. R., Okun, M. S. & Klein, C. Parkinson's disease. *Lancet* **397**, 2284-2303, doi:10.1016/s0140-6736(21)00218-x (2021).
- 10 Dorsey, E. R., Sherer, T., Okun, M. S. & Bloem, B. R. The Emerging Evidence of the Parkinson Pandemic. *J Parkinsons Dis* **8**, S3-s8, doi:10.3233/jpd-181474 (2018).
- 11 Reho, P. *et al.* GRN Mutations Are Associated with Lewy Body Dementia. *Mov Disord* **37**, 1943-1948, doi:10.1002/mds.29144 (2022).

- 12 Chen, X. *et al.* Progranulin does not bind tumor necrosis factor (TNF) receptors and is not a direct regulator of TNF-dependent signaling or bioactivity in immune or neuronal cells. *J Neurosci* **33**, 9202-9213, doi:10.1523/JNEUROSCI.5336-12.2013 (2013).
- 13 Kao, A. W., McKay, A., Singh, P. P., Brunet, A. & Huang, E. J. Progranulin, lysosomal regulation and neurodegenerative disease. *Nat Rev Neurosci* **18**, 325-333, doi:10.1038/nrn.2017.36 (2017).
- 14 Mateo, I. *et al.* Reduced serum progranulin level might be associated with Parkinson's disease risk. *Eur J Neurol* **20**, 1571-1573, doi:10.1111/ene.12090 (2013).
- 15 Kiebertz, K., Katz, R., McGarry, A. & Olanow, C. W. A New Approach to the Development of Disease-Modifying Therapies for PD; Fighting Another Pandemic. *Mov Disord* **36**, 59-63, doi:10.1002/mds.28310 (2021).
- 16 Raffield, L. M. *et al.* Comparison of Proteomic Assessment Methods in Multiple Cohort Studies. *Proteomics* **20**, e1900278, doi:10.1002/pmic.201900278 (2020).
- 17 Bartl, M. *et al.* Blood Markers of Inflammation, Neurodegeneration, and Cardiovascular Risk in Early Parkinson's Disease. *Mov Disord*, doi:10.1002/mds.29257 (2022).
- 18 Abdi, I. Y. *et al.* Cross-sectional proteomic expression in Parkinson's disease-related proteins in drug-naïve patients vs healthy controls with longitudinal clinical follow-up. *Neurobiol Dis*, 105997, doi:10.1016/j.nbd.2023.105997 (2023).
- 19 Hernández, B., Parnell, A. & Pennington, S. R. Why have so few proteomic biomarkers "survived" validation? (Sample size and independent validation considerations). *Proteomics* **14**, 1587-1592, doi:10.1002/pmic.201300377 (2014).
- 20 Füzéry, A. K., Levin, J., Chan, M. M. & Chan, D. W. Translation of proteomic biomarkers into FDA approved cancer diagnostics: issues and challenges. *Clin Proteomics* **10**, 13, doi:10.1186/1559-0275-10-13 (2013).
- 21 Bader, J. M., Albrecht, V. & Mann, M. MS-Based Proteomics of Body Fluids: The End of the Beginning. *Mol Cell Proteomics* **22**, 100577, doi:10.1016/j.mcpro.2023.100577 (2023).
- 22 Pan, C. *et al.* Targeted discovery and validation of plasma biomarkers of Parkinson's disease. *J Proteome Res* **13**, 4535-4545, doi:10.1021/pr500421v (2014).
- 23 Hällqvist, J. *et al.* A Multiplexed Urinary Biomarker Panel Has Potential for Alzheimer's Disease Diagnosis Using Targeted Proteomics and Machine Learning. *Int J Mol Sci* **24**, doi:10.3390/ijms241813758 (2023).
- 24 Captur, G. *et al.* Plasma proteomic signature predicts who will get persistent symptoms following SARS-CoV-2 infection. *EBioMedicine* **85**, 104293, doi:10.1016/j.ebiom.2022.104293 (2022).
- 25 Doykov, I. *et al.* 'The long tail of Covid-19' - The detection of a prolonged inflammatory response after a SARS-CoV-2 infection in asymptomatic and mildly affected patients. *F1000Res* **9**, 1349, doi:10.12688/f1000research.27287.2 (2020).
- 26 Mollenhauer, B. *et al.* Nonmotor and diagnostic findings in subjects with de novo Parkinson disease of the DeNoPa cohort. *Neurology* **81**, 1226-1234, doi:10.1212/WNL.0b013e3182a6cbd5 (2013).

REVIEWER COMMENTS

Reviewer #2 (Remarks to the Author):

The authors have satisfactorily addressed many of the criticisms and suggestions posed by the review panel, and the manuscript is much stronger. However, there are still some issues that warrant clarification.

Please address the following points, which largely relate to terminology and phenonversion issues:

One issue is terminology. The authors continue to focus on the term “Parkinson’s disease” when referring to both PD and to some extent DLB. The coauthors in the recent publication on neuronal a-synuclein disease (NSD) (Simuni et al) settled on that term and abbreviation to encompass the clinical syndromes of PD and DLB under one umbrella term, and they also regarded iRBD as a common prodromal disorder that often eventually manifests as overt PD or DLB. The authors should revise their terminology in various sections of the manuscript to use PD when clearly focused on those with the PD phenotype, and use PD/DLB or NSD or some other term and abbreviation when referring to the combination of these phenotypes or to the same biologic entity with Lewy body disease pathology at its core. And doesn’t Figure 3 reflect “Suggested involved in Lewy body disease” (or NSD) rather than “Parkinson’s disease”

One of several examples of this confusing use of terminology is stated on page 12, “Our approach is feasible to select subjects for clinical trials, as all the iRBD subjects had all of their samples predicted as PD and went on to develop a NSD. Not all iRBD subjects showed the PD pattern. So far, the published data indicates, that around 80% of the iRBD subjects develop NSD.” The term NSD is intended to represent a disease process, and as such, most iRBD subjects represent those who already have the pathobiology of NSD, and many will eventually phenoconvert to the PD or DLB syndrome.

A somewhat related issue is their statements or implications on iRBD almost always phenoconverting to PD. The sets of the largest series on prospectively followed iRBD patients suggest that phenoconversion to PD vs DLB is almost equal, or skewed with PD being slightly more frequent than DLB as eventual phenotypic outcomes. Their plasma proteomic findings are truly intriguing, and may impact the NSD field greatly – including as inclusion criteria or markers worthy of tracking in clinical trials. While more validation work is clearly needed, if their findings are replicated in overt PD and DLB cohorts, and in iRBD subjects who later phenoconvert to either PD or DLB, then the applicability would extend to DLB in addition to PD. Furthermore, their findings would likely NOT be applicable to those with iRBD associated with underlying MSA pathology. Please discuss these issues with more clarity in the Discussion.

Reviewer #3 (Remarks to the Author):

For the revision, the authors do a good job of putting their findings into context and mapping out what is currently possible and what is not and what the requirements are in terms of funding and timescale for follow-ups. I especially like the focus on the data, which – for the discussion of CSF vs plasma – has only limited power to explain things but can classify. I think this is of great benefit to the study.

Overall, my concerns have been addressed. There seems to be a slight technical flaw for Figure S6, so I suggest rerunning this and updating the respective figure(s), which should be a minor fix. Apart from that, my concerns have been addressed.

S6:

In line 85ff of the DNP_Code, feature selection is done on `x_train`. When doing cross-validation (line 161ff), feature selection needs to be done for each `x_train` of the cross-validation split as otherwise there could be some data leakage, as for some splits, datapoints from `x_train` from feature selection could be in `x_test` of cross validation.

There are some additional notes that I would like to share with the authors:

In my initial review, I had pointed to the discrepancy between discovery and target proteins in the study, and the authors now addressed this further and pointed to the general issue of discovery and validation with different instruments. I like the critical discussion about assay technologies. The benefits of using readily available instruments such as the triple quadrupole are evident. However, I would like to point the authors towards ongoing developments. These approaches are characterized as a `triangular` strategies and in clinical proteomics, `rectangular` strategies have been proposed, e.g. see the highly cited review by Geyer et al. (10.15252/msb.20156297) from 2017. Using the same instrument would circumvent this and would enable a much higher consistency between the discovery and target phases.

The newly added Box-Scatter plots give fantastic insight in the data and strengthen the confidence of the findings.

Additionally, the same goes for the heatmap – very insightful. There are some clusters visible, and it could be interesting to explore them further.

The ML model greatly improved, now having additional statistics and increased cross-validation. The code shared on GitHub does not have the *.py-ending, so there is no syntax highlighting. This decreases readability. Additionally, it would be great if the data files could be shared so that the analysis can be reproduced.

As a side note, I found a some figures really hard to read as the font size is too small, especially for the axis. One way to check this is to open Word, set it to 100%, and check what is readable and what is not. Supplementary Figure 6 as bar plots is a bit hard to read; maybe a table with mean + std would be helpful.

In the code: LL 270: # # # Plot precision-recall curve of trainin set -> This is the PR curve of the test set (code is correct but inline comment not)

Reviewer #3 (Remarks on code availability):

The code shared on GitHub does not have the *.py-ending, so there is no syntax highlighting. This decreases readability. The Readme could be extended.

Additionally, it would be great if the data files could be shared so that the analysis can be reproduced.

Reviewer #4 (Remarks to the Author):

This manuscript presents innovative and important findings on peripheral biomarkers of PD, in particular its very early stages. It has been thoroughly responsive to the comments and I don't have anything further to adress.

Rebuttal letter for manuscript NCOMMS-23-14866A

titled

Proteomics and machine learning identify a distinct blood biomarker panel to detect Parkinson's disease up to 7 years before motor disease

We thank the editors and reviewers for their valuable feedback in regard of our manuscript. We have taken on-board all the points, suggestions and the manuscript has been revised accordingly.

Below, we answer all comments point-by-point.

In addition to these point-to-point answers below, we have again involved a native speaker, who revised some grammar and wordings in the text, that is tracked in the files.

Reviewer comments are displayed in black, the author response in blue, quotes from the manuscript in italics.

REVIEWER COMMENTS

Reviewer #2 (Remarks to the Author):

The authors have satisfactorily addressed many of the criticisms and suggestions posed by the review panel, and the manuscript is much stronger. However, there are still some issues that warrant clarification.

Please address the following points, which largely relate to terminology and phenonversion issues:

Reviewer comment: One issue is terminology. The authors continue to focus on the term "Parkinson's disease" when referring to both PD and to some extent DLB. The coauthors in the recent publication on neuronal a-synuclein disease (NSD) (Simuni et al) settled on that term and abbreviation to encompass the clinical syndromes of PD and DLB under one umbrella term, and they also regarded iRBD as a common prodromal disorder that often eventually manifests as overt PD or DLB. The authors should revise their terminology in various sections of the manuscript to use PD when clearly focused on those with the PD phenotype, and use PD/DLB or NSD or some other term and abbreviation when referring to the combination of these phenotypes or to the same biologic entity with Lewy body disease pathology at its core. And doesn't Figure 3 reflect "Suggested involved in Lewy body disease" (or NSD) rather than "Parkinson's disease"

One of several examples of this confusing use of terminology is stated on page 12, "Our approach is feasible to select subjects for clinical trials, as all the iRBD subjects had all of their

samples predicted as PD and went on to develop a NSD. Not all iRBD subjects showed the PD pattern. So far, the published data indicates, that around 80% of the iRBD subjects develop NSD.” The term NSD is intended to represent a disease process, and as such, most iRBD subjects represent those who already have the pathobiology of NSD, and many will eventually phenoconvert to the PD or DLB syndrome.

Our answer: The new Integrated Staging System of Neuronal alpha-Synuclein Disease (NSD-ISS) is indeed an important step, which should be acknowledged. In fact, since we were actively integrated in the development of NSD-ISS we truly appreciate this comment and revised the manuscript accordingly to be compliant with the proposed terminology. Isolated REM sleep behaviour disorder (iRBD) is classified as Stage 2 in the NSD-ISS, Parkinson’s disease (PD) and dementia with Lewy bodies (DLB) is categorized as Stage 3-6 according to the functional impairment of the classified subjects¹.

We now consequently use the terms iRBD, PD, and DLB when we are talking about these clinical syndromes. All these terms are included into the term Neuronal Synuclein Disease (NSD). The transition from the prodromal iRBD stage to a higher stage is named “phenoconversion” from stage 2 to stage 3 of NDS.

We thank the reviewer for the suggestion to also rename **Figure 3**, it has been renamed as “*Suggested involvement in Synuclein Disease*” with the legend: “*Oligomerisation and accumulation of α -synuclein in Lewy body inclusions is a key process in the pathophysiology of Neuronal Synuclein Disease.*”

Here, we list the several changes to the manuscript:

Main manuscript: page 3, lines 106-13: “*Inflammatory risk factors in circulating blood (i.e., C-reactive-protein and Interleukin-6 and α -synuclein-specific T-cells) are associated with motor deterioration and cognitive decline in PD^{2,3}. These inflammatory blood markers can even be identified in plasma/serum samples of individuals with isolated REM sleep behaviour disorder (iRBD), the early stage of a Neuronal Synuclein Disease (NSD), and the most specific predictor for PD and dementia with Lewy bodies (DLB)⁴. NSD was recently proposed as a new biologically defined term, for a spectrum of clinical syndromes, including iRBD, PD, and DLB that follow an integrated clinical staging system of progressing neuronal α -synuclein pathology (NSD-ISS)¹.*”

Main manuscript: pages 3-4, lines 125-48: “*In summary, using a panel of eight blood biomarkers identified in a machine-learning approach, we were able to differentiate between PD and HC with a specificity of 100%, and to identify 79% of the iRBD subjects, up to 7 years before the development of either DLB or motor PD (NSD stage 3). Our identified panel of biomarkers significantly advances NSD research by providing*

potential screening and detection markers for use in the earliest stages of NSD for subject identification/stratification for the upcoming prevention trials.”

Main manuscript: page 7, lines 306-10: “This cohort was available from continuing recruitment at the same centre and consisted of longitudinally followed iRBD subjects. Deep phenotyping revealed 100% (54/54) had RBD on PSG, 88.9% (48/54) had hyposmia as identified with the Sniffin’ Stick Identification Test, and 91.7 % (22/24) had neuronal α -synuclein positivity as shown by α -synuclein Seed Amplification Assay (SAA) in cerebrospinal fluid (CSF)⁵.”

Main manuscript: page 7, lines 324-26: “...at the time of analysis, 16 out of the 54 subjects in our longitudinal iRBD validation cohort had developed PD/DLB.”

Main manuscript: page 9, line 393-95: “In the last years, CSF SAA emerged as the most specific indicator for NSD, including in prodromal stages like iRBD, with an impressively high sensitivity and specificity of up to 74 and 93%, respectively, across various cohorts^{1,6}.”

Main manuscript: page 9, lines 400-2: “Therefore, the identification of additional biomarkers is needed, as is further knowledge of the biomarkers and pathways of the underlying pathophysiology (e.g. inflammation) during the earliest stage of NSD.”

Main manuscript: page 10, lines 456-57: “So far, 16 of the 54 iRBD subjects converted to PD/DLB”

Main manuscript: pages 14, lines 648-55: “Our work was predominantly focused on the similarities between PD and iRBD. Future work would include (i) validation of our findings in independent cohorts consisting of iRBD and other at-risk subjects (e.g. hyposmia), PD, DLB and MSA subjects, (ii) refinement of the panels of biomarkers developed in this study including sensitivity and technical performance, (iii) and using the pipeline described in this manuscript, the identification and validation of additional biomarkers that could distinguish between the different clinical syndromes with the ultimate goal of identifying progression biomarkers as outcome measures for prevention trials.

Main manuscript: pages 13/14, lines 627-47 One advantage of using triple quadrupole platforms is that new and better biomarkers can easily be augmented into the test described in this manuscript. Thus, any test could be refined and optimized over time with very little modification to the assay as new biomarkers are discovered. Clinical testing for neurological disorders is limited to the use of a selected few well-characterised individual markers and translating new biomarkers to eventual clinical application is notoriously challenging. The power of using multiplexed biomarker technologies with machine learning enables biomarkers to be evaluated in context with other markers of pathological events, thereby creating a ‘disease profile’ as opposed to individual markers. This approach opens the biomarker discovery field for many disorders and increases the specificity and sensitivity of testing as demonstrated in this study. The combination of multiplexed analysis of biomarker panels analysed on triple quadrupole platforms can advance biomarker translation to clinical application;

this mass spectral technology is already embedded in many clinical diagnostics labs for routine small molecule analyses."

Supplementary material: page 2, lines 70-3: *"In those subjects who converted to PD/DLB (stage 3 NSD) during follow-up, eight of eleven were observed to have the PD profile through all timepoints and the remaining three patients demonstrated a change to a PD profile during follow-up."*

Supplementary Table S4: page 9. *"Characteristics of longitudinal iRBD subjects (phase II)"**"The table shows age, sex, number of longitudinal samples, and the time since baseline for the last sample. Out of the 40 iRBD subjects, 16 had converted to stage 3 NSD (neuronal synuclein disease) at the time of the last sample (11 Parkinson's disease, 5 dementia with Lewy bodies). The time from baseline to conversion is shown."*

Reviewer comment: A somewhat related issue is their statements or implications on iRBD almost always phenoconverting to PD. The sets of the largest series on prospectively followed iRBD patients suggest that phenoconversion to PD vs DLB is almost equal, or skewed with PD being slightly more frequent than DLB as eventual phenotypic outcomes. Their plasma proteomic findings are truly intriguing, and may impact the NSD field greatly – including as inclusion criteria or markers worthy of tracking in clinical trials. While more validation work is clearly needed, if their findings are replicated in overt PD and DLB cohorts, and in iRBD subjects who later phenoconvert to either PD or DLB, then the applicability would extend to DLB in addition to PD. Furthermore, their findings would likely NOT be applicable to those with iRBD associated with underlying MSA pathology. Please discuss these issues with more clarity in the discussion.

Our answer: We fully agree that the actual literature supports an almost equal conversion rate from iRBD to PD and DLB. Thus, we revised the corresponding parts and included additional references.

Further, we updated the discussion section to discuss our findings in PD and iRBD subjects and the possible next steps for validating the data in DLB and addressing the possible limitations of our data regarding MSA subjects.

Main manuscript: page 13, lines 612-24: *"Continuing further longitudinal follow-up of these subjects will elucidate our understanding of when and potentially why conversion occurs/does not occur. It is known that around 80% of iRBD subjects develop NSD, i.e. PD/DLB with a rate of 6% per year as shown in a multicenter cohort including ours⁹. To a lesser extent, iRBD subjects develop the intracytoplasmic glial α -synuclein aggregation disorder Multiple Systems Atrophy (MSA)^{8,9}. Although RBD is common in MSA (summary prevalence of 73%)¹⁰ none of our iRBD subjects have, as yet, converted to MSA. Recruiting and following large longitudinal at-risk cohorts is, therefore, very important and future studies will not only identify biomarkers for phenoconversion from stage 1 or 2 to eventually stage 3 NSD or MSA, but also identify the many possible factors of resilience (including genetics etc.) of NON-conversion*

which will be as, if not more important than identifying indicators for phenoconversion. Both direction progression biomarkers from stage 1 and 2 cohorts will have tremendous implications for future neuroprevention trials as phenoconversion itself is (due to the low annual rate) unlikely to be an outcome measure."

Reviewer #3 (Remarks to the Author):

For the revision, the authors do a good job of putting their findings into context and mapping out what is currently possible and what is not and what the requirements are in terms of funding and timescale for follow-ups. I especially like the focus on the data, which – for the discussion of CSF vs plasma – has only limited power to explain things but can classify. I think this is of great benefit to the study.

Reviewer comment: Overall, my concerns have been addressed. There seems to be a slight technical flaw for Figure S6, so I suggest rerunning this and updating the respective figure(s), which should be a minor fix. Apart from that, my concerns have been addressed.

Our answer: We gratefully thank the reviewer for pointing this out. The plot has been updated:

Supplementary Figure S6. Classification metrics of the discriminant SVM model, predicting samples as PD or control (phase I)

The classification metrics were calculated from stratified k-fold cross-validation utilising six splits of the data and 40 repetitions and are displayed as histograms showing the frequency of the metrics precision, recall, the F1 score, and the balanced accuracy score. The average and standard deviation was, for precision 0.87 ± 0.09 , for recall 0.87 ± 0.08 , for the F1 score 0.86 ± 0.09 , and for the balanced accuracy score 0.82 ± 0.12 .

Reviewer comment: S6: In line 85ff of the DNP_Code, feature selection is done on x_train. When doing cross-validation (line 161ff), feature selection needs to be done for each x_train of the cross-validation split as otherwise there could be some data leakage, as for some splits, datapoints from x_train from feature selection could be in x_test of cross validation.

Our answer: We thank the reviewer for noting this discrepancy. The cross-validation has been updated to perform feature selection in the training set in each CV split.

Main manuscript: page 6, lines 275-83: *“We further constructed receiver operating characteristic (ROC) and precision-recall (PR) curves to illustrate the ability of each protein to distinguish between PD and HC and compared this with the ability of the combined multiplexed protein panel. The combined panel achieved an AUC of 1.0 on both ROC and PR curves. The AUC of the individual predictors ranged from 0.53 to 0.92 in the ROC curve, and from 0.79 to 0.96 in the PR curve (Figure 2). We further evaluated the whole dataset by performing repeated cross-validation with six splits of the data and 40 repetitions. The resulting classification metrics (Supplementary Figure S6) demonstrated average and standard deviation for precision, recall, F1 score, and balanced accuracy score of 0.87 ± 0.09 , 0.87 ± 0.08 , 0.86 ± 0.09 and 0.82 ± 0.12 , respectively, thereby indicating a highly robust classification model.”*

Reviewer comment: There are some additional notes that I would like to share with the authors:

In my initial review, I had pointed to the discrepancy between discovery and target proteins in the study, and the authors now addressed this further and pointed to the general issue of discovery and validation with different instruments. I like the critical discussion about assay technologies. The benefits of using readily available instruments such as the triple quadrupole are evident. However, I would like to point the authors towards ongoing developments. These approaches are characterized as a `triangular` strategies and in clinical proteomics, `rectangular` strategies have been proposed, e.g. see the highly cited review by Geyer et al. (10.15252/msb.20156297) from 2017. Using the same instrument would circumvent this and would enable a much higher consistency between the discovery and target phases.

The newly added Box-Scatter plots give fantastic insight in the data and strengthen the confidence of the findings.

Additionally, the same goes for the heatmap – very insightful. There are some clusters visible, and it could be interesting to explore them further.

The ML model greatly improved, now having additional statistics and increased cross-validation. The code shared on GitHub does not have the *.py-ending, so there is no syntax highlighting. This decreases readability. Additionally, it would be great if the data files could be shared so that the analysis can be reproduced.

Our answer: The GitHub code has been updated and provided with a.py suffix to increase readability.

https://github.com/jchallqvist/DNP_Pub/blob/main/DNP_Code

Reviewer comment: As a side note, I found some figures really hard to read as the font size is too small, especially for the axis. One way to check this is to open Word, set it to 100%, and check what is readable and what is not. Supplementary Figure 6 as bar plots is a bit hard to read; maybe a table with mean + std would be helpful.

Our answer: The figures have been adjusted with larger font size where possible. In addition, we have added average and standard deviation to the text describing Supplementary Figure 6 as well as to the figure caption.

Reviewer comment: In the code: LL 270: # # # Plot precision-recall curve of trainin set -> This is the PR curve of the test set (code is correct but inline comment not)

Our answer: We thank the reviewer for pointing out this error. We have corrected it.

Reviewer #3 (Remarks on code availability):

Reviewer comment: The code shared on GitHub does not have the *.py-ending, so there is no syntax highlighting. This decreases readability. The Readme could be extended.

Our answer: A.py suffix has been added to increase readability. The Readme has been updated, including information about the package versions, and extended with a statement about data availability.

Reviewer comment: Additionally, it would be great if the data files could be shared so that the analysis can be reproduced.

Our answer: We fully acknowledge that sharing the data files would be beneficial. However, as we are performing additional analyses on the proteomic and clinical data for a grant application, we will keep the data tables embargoed for approximately one year before making them publicly available. We are of course more than happy to share the data tables upon request by e-Mail to the corresponding author.

Reviewer #4 (Remarks to the Author):

This manuscript presents innovative and important findings on peripheral biomarkers of PD, in particular its very early stages. It has been thoroughly responsive to the comments and I don't have anything further to address.

Our answer: We thank reviewer 4 for this comment.

References

- 1 Simuni, T. *et al.* A biological definition of neuronal α -synuclein disease: towards an integrated staging system for research. *The Lancet. Neurology* **23**, 178-190, doi:10.1016/s1474-4422(23)00405-2 (2024).
- 2 Mollenhauer, B. *et al.* Baseline predictors for progression 4 years after Parkinson's disease diagnosis in the De Novo Parkinson Cohort (DeNoPa). *Mov Disord* **34**, 67-77, doi:10.1002/mds.27492 (2019).
- 3 Bartl, M. *et al.* Blood Markers of Inflammation, Neurodegeneration, and Cardiovascular Risk in Early Parkinson's Disease. *Mov Disord*, doi:10.1002/mds.29257 (2022).
- 4 Lindestam Arlehamn, C. S. *et al.* α -Synuclein-specific T cell reactivity is associated with preclinical and early Parkinson's disease. *Nat Commun* **11**, 1875, doi:10.1038/s41467-020-15626-w (2020).
- 5 Concha-Marambio, L. *et al.* Accurate Detection of α -Synuclein Seeds in Cerebrospinal Fluid from Isolated Rapid Eye Movement Sleep Behavior Disorder and Patients with Parkinson's Disease in the DeNovo Parkinson (DeNoPa) Cohort. *Mov Disord* **38**, 567-578, doi:10.1002/mds.29329 (2023).
- 6 Grossauer, A. *et al.* α -Synuclein Seed Amplification Assays in the Diagnosis of Synucleinopathies Using Cerebrospinal Fluid-A Systematic Review and Meta-Analysis. *Mov Disord Clin Pract* **10**, 737-747, doi:10.1002/mdc3.13710 (2023).
- 7 O'Bryant, S. E. *et al.* A proteomic signature for dementia with Lewy bodies. *Alzheimers Dement (Amst)* **11**, 270-276, doi:10.1016/j.dadm.2019.01.006 (2019).
- 8 Zhang, H. *et al.* Risk Factors for Phenoconversion in Rapid Eye Movement Sleep Behavior Disorder. *Annals of neurology* **91**, 404-416, doi:10.1002/ana.26298 (2022).
- 9 Postuma, R. B. *et al.* Risk and predictors of dementia and parkinsonism in idiopathic REM sleep behaviour disorder: a multicentre study. *Brain : a journal of neurology* **142**, 744-759, doi:10.1093/brain/awz030 (2019).
- 10 Palma, J. A. *et al.* Prevalence of REM sleep behavior disorder in multiple system atrophy: a multicenter study and meta-analysis. *Clin Auton Res* **25**, 69-75, doi:10.1007/s10286-015-0279-9 (2015).

REVIEWERS' COMMENTS

Reviewer #2 (Remarks to the Author):

The authors have satisfactorily addressed all of the criticisms and suggestions posed by the review panel.

Reviewer #3 (Remarks to the Author):

Concerns have been adressed.

Reviewer #3 (Remarks on code availability):

The code was now improved.

Small note: The link seems broken and shows a 404, however it points to the right repository and one can fine the code. The correct link would probably be:

https://github.com/jchallqvist/DNP_Pub/tree/main

Rebuttal letter for manuscript NCOMMS-23-14866B

Below, we answer the comments point-by-point.

All the changes to the manuscript are marked in red color in the tracked changes file.

REVIEWERS' COMMENTS

Reviewer #2 (Remarks to the Author):

The authors have satisfactorily addressed all of the criticisms and suggestions posed by the review panel.

Reviewer #3 (Remarks to the Author):

Concerns have been addressed.

Reviewer #3 (Remarks on code availability):

The code was now improved.

Small note: The link seems broken and shows a 404, however it points to the right repository and one can find the code. The correct link would probably be:

https://github.com/jchallqvist/DNP_Pub/tree/main

Our answer:

We thank the reviewers and are happy that we could address all the valuable comments and suggestions.

We checked and corrected all the hyperlinks to the repositories.